# Characterization of fog microphysics and their relationships with visibility at a mountain site in China

Quan Liu[1], Xiaojing Shen[1,*], Junying Sun[1,*], Yangmei Zhang[1], Bing Qi[2], Qianli Ma[3], Lujie Han[3], Honghui Xu[3], Xinyao Hu[1], Jiayuan Lu[1], Shuo Liu[1], Aoyuan Yu[1], Linlin Liang[1], Qian Gao[4], Hong Wang[1], Huizheng Che[1], Xiaoye Zhang[1]

[1] State Key Laboratory of Severe Weather & Key Laboratory of Atmospheric Chemistry of CMA, Chinese Academy of Meteorological Sciences, Beijing, 100081, China

[2] Hangzhou Meteorological Bureau, Hangzhou, 310051, China

[3] Zhejiang Lin'an Atmosphere Background National Observation and Research Station, Hangzhou, 311300, China

[4] Beijing Weather Modification Center, Beijing, 100089, China

*Correspondence to*: Xiaojing Shen (shenxj@cma.gov.cn) and Junying Sun (jysun@cma.gov.cn)

**Abstract.** Enhancing the understanding of fog microphysical processes is essential for reducing uncertainty in fog forecasts, particularly in predicting fog visibility and duration. To investigate the complex interactions between aerosols and fog microphysics and their impacts on visibility degradation, simultaneous measurements of aerosol and fog microphysical characteristics were conducted from April to May, 2023 at a mountain site (1483 m a.s.l.) in the Yangtze River Delta (YRD) region, China. In this study, eight fog events were investigated during the campaign, revealing significantly higher fog droplet number concentrations ($N_d$) compared to those observed in clean areas. A strong correlation was found between pre-fog aerosol number concentration ($N_a$) and the peak $N_d$ of each fog event, indicating the substantial influence of pre-existing aerosol levels on fog microphysics. Water vapor supersaturation ratio (*SS*) within fogs was estimated to 0.07%±0.02%, slightly higher than previous estimates in urban and suburban areas. The broadening of the droplets size distribution (DSD) at stages of formation, development, and mature were dominantly driven by activation, condensation, and collision-coalescence mechanisms, respectively. This evolution process often led DSD to a shift from unimodal to trimodal distribution, with peaks around 6, 12, and 23 μm. For fog events occurring under high $N_a$ background, a notable decrease of temperature during mature stage promoted a secondary activation-dominated process, resulting in the formation of numerous small fog droplets and reducing large droplet size. The evolution of DSD can significantly influence visibility (*VIS*) in fogs. Detailed comparison of several visibility calculation methods suggests that estimating visibility based on the extinction of fog droplets only led to considerable overprediction when 100 m < *VIS* ≤ 1000 m. The results highlight the necessity of incorporating both fog droplet and aerosol extinction in fog visibility forecasts, particularly in anthropogenically polluted regions.

## 1 Introduction

Fog, consisting of suspended liquid droplets or ice crystals near the ground, has substantial impacts on transportation, aviation, and daily activities due to its capability to drastically reduce visibility to less than 1 km (Koračin et al., 2014; Niu et al., 2010a; Gultepe et al., 2015). The formation and types of fog are influenced by various atmospheric conditions and processes. For instance, continental fog commonly forms by radiative cooling of the surface (known as radiation fog) or through the lowering of pre-existing stratus clouds to ground level (Tardif and Rasmussen, 2007). Once the fog forms, its life cycle is influenced by a combination of radiation, turbulence, thermodynamic, and cloud microphysical processes (Mazoyer et al., 2017). These processes interact in complex manners that are not yet fully understood. Advancing the understanding of fog microphysical processes is essential for improving fog forecasts (Boutle et al., 2015; Martinet et al., 2020), particularly in predicting the timing of fog formation and dissipation (Van Der Velde et al., 2010; Boutle et al., 2018).

The interactions between aerosol particles and fog droplets are complicated (Fan et al., 2016). The fog processes can scavenge large amounts of aerosols, altering their chemical composition, size distribution, and mixing state (Schroder et al., 2015; Roth et al., 2016; Qian et al., 2023). Conversely, Aerosol particles can serve as cloud condensation nuclei (CCN) in supersaturated water vapor environments (Twomey, 1959), playing an important role in the evolution of fog. The concentration, size distribution, and chemical composition of aerosols can significantly influence fog microphysical characteristics and optical properties (Dusek et al., 2006; Zhao and Garrett, 2015; Zhang et al., 2024). For example, in regions with intense anthropogenic activities, the abundance of CCN can lead to the formation of numerous but smaller fog droplets (Li et al., 2017; Twomey, 1977) and prolonging fog atmospheric lifetime (Yan et al., 2020; Jia et al., 2019). This can enhance the light scattering, thereby reducing visibility more effectively than that in cleaner environments with fewer but larger droplets. Additionally, the activation capacity of aerosol particles is mainly determined by their size distribution and chemical composition (Andreae and Rosenfeld, 2008; Gysel et al., 2007). Particles with high activation capacity can lower the critical activation supersaturation threshold needed for droplet formation (Ervens et al., 2005; Zhang et al., 2012; Wang et al., 2024). Therefore, the influence of aerosols on fog microphysics varies across regions with different aerosol backgrounds.

Given that visibility degradation is the most significant hazard during fog events, accurately estimating visibility is crucial for fog prediction. Numerous previous studies have focused on the relationship between fog microphysical parameters and visibility. Eldridge (1961) identified a strong negative correlation between fog visibility ($VIS$) and liquid water content ($LWC$) based on fog observations (Eldridge, 1961). In addition to $LWC$, Meyer et al. (1980) suggested that there is a significant negative correlation between fog $VIS$ and droplet number concentration ($N_d$). Kunkel (1984) suggested that $LWC$ could serve as the single parameter for visibility parameterization for fog, based on observation data of 11 fog cases. To improve fog visibility predictions, a dual-parameters scheme ($LWC \cdot N_d$), relating both of $LWC$ and $N_d$ to $VIS$, was proposed and optimized by Gultepe et al. (2006). This dual-parameters scheme demonstrated higher forecast accuracy compared to the $LWC$-only scheme (Zhang et al., 2014). Furthermore, Song et al. (2019) suggested that $VIS$ is not only related to $LWC \cdot N_d$ but also to the effective diameter ($D_{eff}$) of droplet size spectrum. They incorporated $D_{eff}$ into the dual-parameter scheme based on

fog observations in the mountainous regions of Korea. However, the fitting parameters in these parameterization schemes are influenced by the characteristics of fog DSD, and their values vary significantly in different regions and environments (Kunkel, 1984; Gultepe and Milbrandt, 2007; Zhang et al., 2014). Such variability of these parameters emphasizes the strong regional dependence of the applicability of these two parameterization schemes. Additionally, Zhang et al. (2014) examined these parameterization methods using in situ measurement data from four fog cases in a region of intense anthropogenic

emissions, and they found that these parameterizations were unsuitable for light fog events. This is caused by only the extinction caused by fog droplets is taken into account in these fog visibility parameterization schemes. The extinction contribution from hygroscopic growth of unactivated aerosol particles under water vapor supersaturation conditions may not be ignored (Elias et al., 2009; Hammer et al., 2014). However, few studies have utilized simultaneous microphysical observations of fog droplets and aerosols to evaluate their contributions to visibility during fog evolution.

To improve the understanding of the interactions between aerosols and fog microphysics and their impacts on visibility degradation in polluted regions, simultaneous measurements of number size distributions of aerosol particle and fog droplet were conducted at a mountain site in the megacity cluster of the YRD region, China. In this study, eight fog events are discussed in detail to illustrate the potential impacts of different aerosol concentration background on fog microphysical characteristics. Details on the observation site, instrumentation, sampling inlet system for fog interstitial particles and fog

residual particles, and the *SS* estimation methods are described in the Measurement and methodology section. In the Results and discussions section, we first present general observations during this campaign in Section 3.1 and discuss the relationship between pre-fog aerosols and fog droplets in Section 3.2. Then, the variations of *SS* values derived by aerosol and fog measurements are presented in Section 3.3. The temporal evolution of fog DSD for two typical fog events is characterized and discussed in Section 3.4. Finally, the contributions of aerosols and droplets to visibility during different

stages of fog evolution are presented in Section 3.5. The summaries are provided in the Conclusions and implications section.

## 2 Measurement and methodology

### 2.1 Observation site

Simultaneous measurements of aerosol particle number size distribution (PNSD), CCN number concentration, and fog microphysical parameters ($N_d$, $LWC$, $D_{eff}$) were conducted during April 11th to May 8th, 2023 at the summit of Mt. Daming in

Hangzhou, China. The mountain site (30.03°N, 119.00°E, 1483 m a.s.l.) locates in the southwest Hangzhou area with a distance of ~120 km from Hangzhou downtown (Fig. S1), belonging to the YRD region. The surroundings of this site have no distinct anthropogenic emissions apart from a few villages at the base of the mountain. Due to the unique geography, the site frequently experiences various cloud/fog events, such as orographic cloud, radiation fog, and stratus-lowering fog.

## 2.2 Instrumentation and Methods

### 2.2.1 Sampling inlet system

To simultaneously measure the physicochemical properties of fog interstitial particles and fog residual particles, an automatic three-way switching inlet system was developed, incorporating a PM$_{2.5}$ cyclone and a Ground-based Counterflow Virtual Impactor (GCVI) (model 1205, Brechtel Manufacturing Inc., USA) (Fig. S2). This system utilized two electromagnetic ball valves installed downstream of the PM$_{2.5}$ cyclone and GCVI pathway, respectively, and was controlled by a custom LabView (National Instruments, Austin, USA) software. The inlet system was installed on the roof, approximately 5 m above the ground. The aerosol measurements were performed downstream of this inlet system, including PNSD, CCN concentration at different water vapor saturation (*SS*), and aerosol chemical composition. The three-way valve switching is controlled automatically based on fog and fog-free conditions. Fog condition was detected by using visibility and RH sensors integrated into the GCVI system, with thresholds set at 1000 m for visibility and 95% for RH. Under fog-free conditions, ambient air was sampled through the PM$_{2.5}$ inlet and dried by an automatic regenerating absorption aerosol dryer, ensuring the relative humidity (RH) in the sample flow remained below 30% (Tuch et al., 2009). Under fog conditions, the sampling system alternated between the PM$_{2.5}$ cyclone and GCVI pathways every 30 minutes. During fog events, particles collected through the PM$_{2.5}$ cyclone pathway represent fog interstitial particles, while particles sampled and dried via the GCVI pathway represent fog droplet residual particles.

The GCVI system uses a compact wind tunnel placed upstream of the CVI inlet (model 1204) to accelerate cloud/fog droplets into the CVI inlet tip. Droplets smaller than the cut size of CVI inlet are rejected from the tip by the counterflow. Droplets larger than the cut size but smaller than the maximum size limit pass through the tip and are dried into small residue particles. For a given counterflow, airspeed within the wind tunnel, temperature, and pressure, the cut size of droplet that penetrates into the inlet is fixed. In this study, the GCVI inlet sampled droplets with aerodynamic diameters larger than 7.8 μm by setting the airspeed and counter flow to 90 m s$^{-1}$ and 4 L min$^{-1}$, respectively. The droplets were then dried using an evaporation chamber (airflow temperature at 40 ℃) in the GCVI. Details of the GCVI system can be found in other studies (Shingler et al., 2012; Bi et al., 2016; Karlsson et al., 2021). It is worth noting that the GCVI tends to yield a higher number concentration of cloud particles compared to the actual ambient cloud particle concentration, which should be corrected using an enrichment factor (EF). The EF was calculated based on the GCVI sampling flow settings, airspeed, and its geometry configuration, as recommended by Shingler et al. (2012). In this work, an EF of 5.9 was derived for airspeed of 90 m s$^{-1}$. Therefore, the concentration measured at the downstream of the GCVI pathway has been corrected by the EF of 5.9.

### 2.2.2 Fog microphysical parameters

A Fog Monitor (Model FM-100, DMT Inc., USA) was applied in situ for measuring real-time droplets size distribution (DSD) within the size range of 2-50 μm. The inlet of FM-100 sampled air approximately 2.5 m above the ground. Droplets are sorted into the 20 predefined size bins with a measuring time-resolution of 1 s. The values of fog microphysical

parameters ($N_d$, *LWC*, and $D_{eff}$) were calculated from fog DSD according to the equations addressed by Spiegel et al. (2012):

$$N_d = \sum N_i \quad (1)$$

$$LWC = \frac{\pi}{6} \sum N_i D_i^3 \rho_w \quad (2)$$

$$D_{eff} = \sum N_i D_i^3 \, / \, \sum N_i D_i^2 \quad (3)$$

where $N_i$ is the droplet number concentration in the *i*th bin, $D_i$ denotes the diameter in the *i*th bin, and $\rho_w = 1$ g cm$^{-3}$ represents for the density of pure water.

### 2.2.3 Aerosol measurements

The dry PNSDs were measured by a Twin Scanning Mobility Particle Sizer (TSMPS, TROPOS, Germany), consisting of a Differential Mobility Analyzer (DMA) and a Condensation Particle Counter (CPC, Model 3772, TSI Inc., USA). The TSMPS system measured the PNSD within the range 10-850 nm in mobility diameter with an X-ray neutralizer. Each scan was set to 5 min for every loop with a total sample flowrate of 2.5 L min$^{-1}$.

The CCN number concentration ($N_{CCN}$) was measured at various *SS* using a Cloud Condensation Nuclei Counter (Model CCN-100, DMT Inc., USA). In this study, the CCN counter was sequentially set to four supersaturation (*SS*) values: 0.1%, 0.2%, 0.4%, and 0.7%, each for a duration of 5 minutes. The four *SS* setpoints were sequentially scanned from low to high and then back from high to low to avoid large change of SS in the CCNc column. Due to the cloud chamber inside the CCN counter requires time to stabilize the temperature after each change in *SS*, data measured in the first minute of each *SS* were excluded. The ratio of sample flow and sheath flow was set at 1:10, with the flowrate of 0.45 L min$^{-1}$ and 4.5 L min$^{-1}$, respectively. The *SS* calibration of CCNc-100 was performed with ammonium sulfate particles before and after the campaign.

Aerosol chemical components were measured by a high-resolution time-of-flight aerosol mass spectrometer (HR-ToF-AMS, Aerodyne Inc., USA) (Canagaratna et al., 2007; Decarlo et al., 2006), including nitrate, sulfate, chloride, ammonium, and organics. Black carbon (BC) mass concentrations were obtained by using a single particle soot photometer (SP2, DMT Inc., USA) (Schwarz et al., 2006; Liu et al., 2020a). The aerosol chemical compositions in this study were used to derive their hygroscopic parameter ($\kappa$) following the method by Liu et al. (2023). The $\kappa$ value for each pure chemical species is provided in Table S1. Detailed analysis on chemical properties of cloud interstitial particles and droplet residual particles will be presented in a subsequent study.

### 2.2.4 Fog event selection criteria

The definition of fog event in this study requires the following conditions to be met simultaneously: visibility less than 1000 m, relative humidity greater than 95%, and fog droplet number concentration greater than 10 cm$^{-3}$ (Lu et al., 2013; Deng et al., 2009; World Meteorological Organization, 2017). Intervals between fog events need to include at least three consecutive hours of fog-free period. In order to avoid precipitation interference in fog measurements, those processes in

which fog appeared after precipitation were eliminated from the later analysis. Hereby, there were 8 available fog events in total were selected to analyze in following text. The detail description for the eight fog events was summarized in Table 1.

### 2.2.5 Method to estimate the *SS* in fog

The *SS* in fogs, as one of the most important environmental parameters in response to fog evolution, cannot be directly measured. Aerosol particles will be activated when their critical activation *SS* lower than the maximum *SS* value of ambient air. In return, cloud/fog droplets can be formed by those particles whose diameters exceed the critical activation diameter ($D_c$) corresponding to that critical activation *SS*. In order to illustrate the influences of *SS* evolution on droplets size distribution, we used two approaches to derive *SS* (Fig. 1). In the first approach, the averaged pre-fog PNSD represented the aerosol background before activation occurrence. The $D_c$ here was determined as the particle size at which the $N_d$ equaled to the integrated aerosol concentration of the pre-fog PNSD from the upper limit down to $D_c$ (Fig. 1a). Then, the corresponding *SS* ($SS_{PNSD}$) was calculated by using the $\kappa$-Köhler equation (Petters and Kreidenweis, 2008) with an averaged $\kappa$ of pre-fog aerosols. In the second approach, the $N_d$ in the fog can be considered to be consistent with the activated CCN number concentration ($N_{CCN}$). The *SS* ($SS_{CCN}$) was determined as the $N_d$ is equivalent to $N_{CCN}$ by using piecewise linear interpolation of the pre-fog *SS*-resolved $N_{CCN}$ measurements (Fig. 1b). Due to the lowest *SS* setpoint in this study is 0.1%, *SS* values less than 0.1% were estimated from extrapolation of linear extension line (magenta dashed line in Fig. 1b).

### 2.2.6 Visibility measurement and calculation

The extinction coefficient of aerosol particles and fog droplets can be calculated from their number size distribution, respectively, according to following equation:

$$b_{ext} = \int Q_{ext} \frac{\pi}{4} D_i^2 N_i(D_i) dD_i \qquad (4)$$

Where $b_{ext}$ is the extinction coefficient, $Q_{ext}$ is the extinction cross section calculated by the droplet (or aerosol particle) diameter ($D_i$) and wavelength of light (880 nm, consistent with the visibility meter) using the Mie theory. The refractive indices of pure composition relevant to the Mie calculations are provided in Table S1. Then, the extinction coefficient is converted to *VIS* using an equation given by [Koschmieder, 1924] as:

$$VIS_{cal} = -\frac{ln\ \varepsilon}{b_{ext}} \qquad (5)$$

where $\varepsilon$ is the brightness contrast threshold. The visibility was also simultaneously measured by a forward scattering visibility meter (Model DNQ1, Huayun Inc., China) at 880 nm, with the range of 0.01-35 km. To make the $VIS_{cal}$ is comparable with the measured *VIS*, the $\varepsilon$ value here is set to 0.05, which is in accordance with the method of visibility meter.

### 2.2.7 Parameterization schemes of fog visibility

Previous studies have explored the relationship between fog microphysical parameters (i.e., *LWC*, $N_d$, and $D_{eff}$) and

visibility (Gultepe et al., 2006; Song et al., 2019; Kunkel, 1984). A commonly used approach for estimating fog visibility was proposed by Kunkel (1984) as follow:

$$VIS_K = \frac{a}{LWC^b} , a = 0.027, b = 0.88 \quad (6)$$

This parameterization scheme is based only on *LWC* and is therefore widely applied in numerical models. However, the parameter of $N_d$ can also significantly influence fog visibility. On this basis, the parameterization was developed by Gultepe et al. (2006) with utilizing both *LWC* and $N_d$:

$$VIS_G = \frac{c}{(LWC \cdot N_d)^d} , c = 1.002, d = 0.65 \quad (7)$$

## 3 Results and discussions

### 3.1 Overview of the observation

Fig. 2 shows the temporal variations of meteorological parameters, cloud microphysical parameters, and aerosol size distribution measured in the field observation from April 11th to May 8th, during which the 8 available fog events are observed. The temperature was above 0 ℃ during the entire observation period, indicating all of the observed fogs were warm fog processes. The wind speed and direction are shown by a polar plot of in Fig. S3. The prevailing wind direction throughout the study period was westerly, with strong winds (exceeding 8 m/s) primarily originating from the west and southwest. In contrast, during foggy periods, the prevailing wind direction shifted to the northeast, with the main wind speed ranging from 4 to 8 m/s. The visibility variations at this site exhibited distinct characteristics, with values predominantly concentrated in high and low ranges (Fig. 2b), without the gradual increase or decrease typically observed in urban areas (Qiang et al., 2015; Wang et al., 2015). Moreover, When RH < 75%, the visibility remained above 10 km, whereas it declined below 1 km when RH > 95%. This indicated that low-visibility events at the site were predominantly driven by fog processes during the observation period.

Large ranges of fog microphysical parameters were observed during the campaign. The median values of $N_d$, *LWC* and $D_{eff}$ of the 8 fog events varied over the ranges of 146–834 cm$^{-3}$, 0.009–0.216 g m$^{-3}$, and 5.5–12.2 μm, respectively (Table 1). The concentration levels of fog droplets varied by orders of magnitude in different environments, ranging from tens in marine and remote background environments (Duplessis et al., 2021; Gultepe et al., 2009) to hundreds in anthropogenically polluted environments (Li et al., 2020; Shen et al., 2018). The variations of $N_d$ and *LWC* showed a consistent trend during fog formation and dissipation stages. However, after fog formation, the trends of the two variables may diverge (Fig. 2c), which is closely related to the variations in $D_{eff}$ (Fig. 2d). The relationship between $N_d$ and *LWC* during the 8 available fog events is presented in Fig. S4 to further illustrate their correlation. There appears to be no obvious correlation between the overall $N_d$ and *LWC*. However, when binning $N_d$ and *LWC* according to $D_{eff}$ values, a notable high linear correlation showed up. This result indicates that using a single parameter to describe cloud microphysical properties may introduce significant uncertainty, which will be further discussed in detail in Section 3.5. For a given range of *LWC* values, $N_d$ generally decreases

as $D_{eff}$ increases. This negative correlation between them is ubiquitous in fog, as the presence of more droplets competes for available water vapor, thereby inhibiting their growth (Li et al., 2017).

Although there were few anthropogenic sources near the site, the observed aerosol concentrations varied dramatically. As shown in Fig. 2e, the $N_a$ ranged from 230 to 15620 cm$^{-3}$, with a median of 2750 cm$^{-3}$. Episodes with $N_a$ exceeding 8000 cm$^{-3}$ were typically associated with a pronounced increase in aerosol number concentration within the size range of 100-100 nm (Fig. 2e), which were likely driven by new particle formation (Shen et al., 2022). In the subsequent discussion, the pre-fog aerosol concentration below and above this median were defined as low and high number concentrations of aerosol backgrounds, respectively.

## 3.2 Relationship between pre-fog aerosols and fog droplets

Previous studies suggested the maximum $N_d$ during cloud/fog formation period was not only depended on the $SS$ reached by the air mass (Mazoyer et al., 2019; Pruppacher and Klett, 2010), but also had a high correlation between the pre-fog or cloud base aerosol number concentrations ($N_a$) (Duplessis et al., 2021; Hegg et al., 2012). Pre-fog $N_a$ here was defined as the average of the last hour before fog formation. As it shown in Fig. 3a, the pre-fog $N_{a\_total}$ (integrated concentration from PNSD measured by TSMPS) had a high correlation with the peak $N_d$ for these fog events, indicating the peak $N_d$ was significantly influenced by pre-fog aerosol. Although there is a temporal difference between the observation of pre-fog aerosols and the subsequent fog process at a fixed site, the measured pre-fog aerosol particles may not fully represent the particles that actually activated into fog droplets. However, due to the high altitude of this mountain site, it is located above the top of the boundary layer for most of the day (Sun et al., 2018). The aerosol physicochemical properties at this altitude are relatively homogeneous and regionally representative, resulting in a good correlation appeared between the pre-fog aerosol and the peak $N_d$. Conversely, the good correlation between them also indicated the observations at this site were representative of a relatively large spatial scale. This provides a rational basis for estimating water vapor supersaturation by using the pre-fog aerosol size distribution in Section 3.3. For the fog events occurred after precipitation (hollow cycles in Fig. 3), the pre-fog $N_a$ and did not follow this linear relationship. This further supports that such processes should be removed from analysis of aerosol effects on fog microphysics.

The linear fitting slopes in Fig. 3, primarily depending on aerosol chemical composition and size distribution, can be used to associated with activation ratio of bulk aerosol. The slope value of 0.09 in this study is significantly higher than the 0.014 observed by Duplessis et al. (2021) on the eastern coast of Canada, indicating stronger bulk activity observed at this mountain site. The difference in the slope can be attributed to both different aerosol properties and $SS$ conditions in the studies. The comparison of $SS$ in various observation environments will be discussed in Section 3.3. In addition, the concentrations of particle diameter larger than 100 nm ($N_{a\_100}$) or 70 nm ($N_{a\_70}$) had a much stronger correlation with the peak $N_d$ than that of total pre-fog $N_a$ (Fig. S5 and Fig. 3b). Previous studies have reported that the peak $SS$ estimated in fogs are typically low (0.03–0.05%) (Mazoyer et al., 2019; Shen et al., 2018), indicating particles with a size smaller than 70 nm

should not be activated in foggy conditions. The result suggests that a proper selection of particle size range is crucial for estimating the peak $N_d$ by using pre-fog $N_a$.

### 3.3 Estimating water vapor supersaturation in fog

The time series of $SS_{PNSD}$ and $SS_{CCN}$ derived from above two approaches (mentioned in Section 2.2.5) during a typical fog event (E3) are shown in Fig. 4a. Although their temporal variations exhibit a high consistence, the mean value of $SS_{PNSD}$ is approximately 30% higher than $SS_{CCN}$. Because of most $SS_{CCN}$ values lower than the lowest $SS$ setpoint (0.1%), substantial uncertainties were introduced by linear extrapolation when deriving $SS_{CCN}$. Therefore, the variations of $SS_{PNSD}$ were considered to be closer to the actual situation and were used in subsequent discussions with a brief symbol of $SS$. Note that, the $SS$ estimation here only considered adiabatic processes such as activation and condensation, and ignores non-adiabatic processes such as collision-coalescence (Wang et al., 2021). If the reduction of $N_d$ caused by the collision-coalescence process is considered, the actual effective $SS$ should be greater than the calculated value.

After fog formation, the $SS$ had a strong negative correlation ($r$=-0.85, $p$<0.001) with ambient temperature (Fig. 4a and Fig. S6), indicating the decrease of temperature played a critical role in supplying sufficient $SS$ for particles activation. Due to incomplete observation data of PNSD or DSD for several fog events during this campaign, here only five events with complete data of the entire process were available for the SS statistics (Fig. 4b). The median $SS$ values for each fog event varied in the range of 0.05%-0.13%, and the 95[th] quantile values were generally less than 0.1% except for the E4. During the whole observation period, the $SS$ varied between 0.01% and 0.25%, with an average of (0.07±0.02)%. This is slightly higher than the fog $SS$ reported in urban 0.05% (max. 0.05%, Shen et al., 2018), suburb with a median of 0.043% (median 0.043%, Mazoyer et al., 2019), and coast (average 0.037%, Duplessis et al., 2021) environments, but significantly lower than that derived from aircraft measurements of clouds (0.10%-0.50%, Gong et al., 2023). The estimated $SS$ in various observation environments seems to be positively correlated with altitude. This can be partly attributed to the lower aerosol number concentration and temperature at high altitudes (Liu et al., 2020b), which reduce excess water vapor consumption in clouds or fog, as well as the equilibrium water vapor pressure (Baccarini et al., 2020; Shen et al., 2018), thereby promoting supersaturation.

### 3.4 Temporal evolution of fog DSD

To explore the temporal evolution of fog, it is common to divide the process into various stages based on changes in visibility (Mazoyer et al., 2022; Niu et al., 2010b; Pilie et al., 1975). Upon this, each fog event in this study was divided into four stages, determined by the changes in visibility computed using a 15-minute running average (refer to the color-time divisions in Fig. 5a). In the formation stage (blue line), there was a pronounced decline in visibility from 1000 to 100 meters within 20 minutes for all cases. In the development stage (magenta line), the visibility continued to decrease but at a significantly slower rate until reaching its minimum value. During the mature stage (brown line), the visibility undergone a slight increase or remains stable. Finally, during the dissipation stage (purple line), the visibility increased rapidly to 1000

meters. As we know, in-situ observations at a fixed site face significant challenges in continuously measuring the evolution of aerosols and fog droplets within a specific air mass. Here, we assume that at a certain height within the fog, the aerosols and fog droplets exhibit similar microphysical characteristics and undergo synchronous variations. Therefore, during a fog process, measurements at different time points at this site can, to some extent, reflect the evolution of the microphysical characteristics of aerosols and cloud droplets at that height.

As it shown in Fig. 5, two typical fog events, characterized by low and high pre-fog aerosol concentration conditions, were selected and analyzed in terms of the evolution of their microphysical characteristics. The averaged fog DSD during various stages is shown in Fig. 6 and Fig. S7. The similar information for the other three fog events was presented in Fig. S8 and S9. Under low aerosol concentration background (E2), as the supersaturation ratio increases in the formation stage, $N_d$ rapidly reached a peak within a short period, while both $LWC$ and $D_{eff}$ exhibited slow growth (Fig. 5c). This indicated that the fog droplets in this stage primarily formed through aerosol particle activation processes, which yielded small droplets with diameters less than 6 μm (Fig. 6a). During the development stage, the $N_d$ continued to increase due to persistent activation of aerosol particles, along with both $LWC$ and $D_{eff}$ gradually increased to their maximum values. Another peak in the fog DSD emerges around 12 μm in this stage (Fig. 6a), indicating that the condensation process began to dominate the broadening of the DSD. In the subsequent mature stage, $N_d$ experienced a significant decrease due to a substantial reduction in small droplets, then maintained a relatively stable value. This indicated that the excess water vapor, defined as the difference of the ambient water vapor pressure and the equilibrium value, was produced and consumed in approximate balance, thus reaching a quasi-stationary supersaturation state. Compared to the development stage, $D_{eff}$ notably increased at this stage, with the main peak of the DSD shifted from 12 μm to 15 μm and an additional considerable peak appearing at 23 μm (Fig. 6a). These changes in fog microphysical characteristics suggest the occurrence of collision-coalescence process, leading to further broadening of the DSD towards larger sizes. After triggering the collision-coalescence mechanism, apart from small fog droplets, certain un-activated aerosol particles were scavenged by the uptake of larger fog droplets. This can be supported by variations of the activation ratio (AR) of cloud residual particles. The AR here was defined as the CCN number concentration measured by the CCNc relative to the total particle concentration (10-850 nm). If fog residual particles enter droplet though an activation process, these particles should also be activated in the CCNc column, where can set different SS conditions. Therefore, the concentrations measured by CCNc and TSMPS downstream of the GCVI inlet should be consistent, i.e., the AR should be approximate 1, especially for high SS setpoints. Fig. 7 shows the variation of AR with $D_{eff}$ at $SS = 0.2\%$, while the results for other $SS$ setpoints are provide in Fig. S10. As it shown, the AR measured downstream of the GCVI airflow were closed to 1 when the $D_{eff}$ smaller than 12 μm. However, when the collision-coalescence process occurred, indicated by $D_{eff}$ exceeding 12 μm (Fig. 5d), the AR of fog residual particles notably decreased. The reduced AR of fog residual particles was caused by the uptake of particles less prone to activation into droplets, implying the removal efficiency for these particles significantly enhanced in this stage. Besides that, both $SS$ and $LWC$ fluctuate around a stable value in the mature stage (Fig. 5c), indicating that evaporation and condensation of water vapor were in a quasi-equilibrium state. In the dissipation stage, $N_d$ and $LWC$ decline rapidly to zero, with a gradual disappearance of droplets in the DSD from

315 large to small sizes (Fig. 6a).

Under high aerosol concentration background (E3, in Fig. 5), the evolutions of fog microphysical characteristics during the formation and development stages were generally consistent with those in E2. However, after reaching and maintaining a quasi-stationary supersaturation state ($SS_{Q1}$) in the early mature stage, a notable decrease in temperature occurred (Fig. 5a) without obvious changes in wind direction and speed (Fig. 5b). This decrease caused an increase in both excess water vapor
pressure and supersaturation, as the temperature-dependent equilibrium vapor pressure dropped faster than the ambient partial vapor pressure. Consequently, a new quasi-stationary supersaturation state ($SS_{Q2}$) was established, exhibiting distinct fog microphysical characteristics (Fig. 6b). Compared to $SS_{Q1}$, the $N_d$ substantially increased in the $SS_{Q2}$ stage, while the $LWC$ and $D_{eff}$ notably decreased (Fig. 5c). The enhanced $SS$ facilitated the further activation of smaller particles that were un-activated during the $SS_{Q1}$ stage, resulting in a secondary activation-dominated process during the E3 (Fig. 5d and Fig. 6b).
During this secondary activation process, a greater number of small droplets formed and competed for the limited water vapor, which led to a decrease in the $D_{eff}$ (Fig. 6b).

### 3.5 Links between fog microphysical parameters and visibility

### 3.5.1 Comparison of different fog visibility estimating methods

Given that visibility degradation is the primary hazard during fog events, establishing an appropriate visibility
parameterization scheme in fog is crucial for improving the accuracy of fog visibility forecasts. Compared to the parameterization schemes of fog visibility, Mie theory incorporates a specific extinction algorithm based on physical processes. Therefore, the fog visibility derived from fog DSD and Mie theory ($VIS_{DSD}$) is expected to better reflect actual conditions, which can serve as a reference for fog visibility parameterization. In this study, we re-established the parameters $a$, $b$, $c$, and $d$ in Equation 6-7 using our measured data. The reconstructed visibility calculations were denoted as $VIS_{KN}$ for
the $LWC$-only parameterization and $VIS_{GN}$ for the $LWC \cdot N_d$ parameterization. Fig. 8a presents a comparison of the calculated visibility based on different parameterization schemes with $VIS_{DSD}$. Compared to $VIS_K$ and $VIS_G$, the deviations of $VIS_{KN}$ and $VIS_{GN}$ from $VIS_{DSD}$ are significantly reduced, especially for $VIS_{GN}$, which has a linear fitting slope of 1.1. This indicates that the dual-parameters scheme of $LWC \cdot N_d$ can better describes visibility degradation contributed by fog droplets. The visibility degradation contributed by fog droplets is determined by fog droplets size distribution. Meanwhile, the fog microphysical
parameters of $N_d$, $LWC$, and $D_{eff}$ are derived from the measurement of fog droplets size distribution (Equation 1-3). When both $LWC$ and $N_d$ values are given, the information of $D_{eff}$ can also be determined (Fig. S4). Comparing to the $LWC$-only parameterization, the $LWC \cdot N_d$ parameterization can better characterize the fog droplets size distribution, and therefore is expected to be more accurate in fog visibility forecasts.

To further evaluate the applicability of the $VIS$ calculation methods mentioned above, we compared these calculated
results with the visibility measured by a visibility meter (Fig. 8b). The $LWC$ and $VIS$ exhibited an exponential relationship, with an inflection point appearing at ~100 m (Fig. S11). Accordingly, the relevant data were analyzed by dividing them into

two intervals: $VIS_{obs} \leq 100$ m and $100$ m $< VIS_{obs} \leq 1000$ m. The results showed that the visibility calculation methods used in Fig. 8b tended to be slightly overestimated to different degrees, with the linear fit slopes being 1.33 for $VIS_{KN}$, 1.16 for $VIS_{GN}$, and 1.21 for $VIS_{DSD}$. The dual-parameters method of $LWC \cdot N_d$ yielded a smaller deviation than that of the $LWC$-only method. However, when $100$ m $< VIS_{obs} \leq 1000$ m, the $VIS$ calculated from the three methods were substantially higher than $VIS_{obs}$, with no obvious correlations between them. This large difference was induced by the visibility data used for the development of visibility parameterizations relied on Mie calculations rather than observed results from a visibility sensor. Additionally, the parameterization schemes in those studies were derived from observations in relatively clean areas, where visibility degradation is predominantly caused by fog droplets. However, these schemes would induce in large uncertainties in visibility calculations in polluted areas, such as the North China Plain (Zhang et al., 2014), where aerosol concentration and extinction contribution can be much higher, especially in light fogs.

### 3.5.2 Aerosol effects on estimating fog visibility

To quantitatively estimate the aerosol contribution on visibility degradation in fog, the dry PNSD of cloud/fog interstitial particles was used to calculate their extinction. Due to the lack of aerosol particle hygroscopic growth factor of aerosol particles under supersaturated conditions, a rough estimation method was proposed to convert dry PNSD to ambient PNSD. This method was based on the continuity of the PNSD and assumed that the maximum dry particle size of fog interstitial particles, after hygroscopic growth under supersaturated conditions, corresponded to the cut-size of the sampling inlet ($PM_{2.5}$ cyclone). It also assumed that the hygroscopic growth factor was constant across different particle sizes. Based on these assumptions, the hygroscopic growth factor of particles under supersaturated conditions can be obtained (Fig. S12). Then the ambient aerosol contribution on visibility can be calculated based on Mie theory.

Fig. 9 shows the comparison of visibility estimation based on only fog droplets and both fog droplets and interstitial particles, respectively. When $VIS_{obs} \leq 100$ m, the high concentration and large size of the fog droplets dominate the visibility degradation. In this situation, the extinction effect of aerosols can be neglected. However, when $100$ m $< VIS_{obs} \leq 1000$ m, estimating visibility based on only fog droplet extinction led to substantial deviations, whereas considering both fog droplet and aerosol extinction significantly reduced the discrepancy between calculated and observed $VIS$. The comparison highlights the importance of considering both fog droplet and aerosol extinction in visibility forecasting during light fog conditions, particularly in polluted regions affected by anthropogenic emissions.

### 4 Conclusions and implications

To explore interactions between aerosols and fog microphysics and their impacts on visibility degradation, this study conducted simultaneous measurements of aerosol and fog microphysical characteristics in spring 2023 at the summit of Mt. Daming (1483 m), located in the YRD region, China. During this campaign, 8 fog events were observed. The median values of $N_d$, $LWC$, and $D_{eff}$ for the 8 fog events varied within the ranges of 146–834 cm$^{-3}$, 0.009–0.216 g m$^{-3}$, and 5.5–12.2 μm,

respectively. A strong correlation was found between pre-fog $N_a$ and the peak $N_d$ of each fog event, implying the potential influence of pre-existing aerosol levels on fog microphysics. Two approaches for deriving $SS$ within fogs were proposed, based on measurements of PNSD and $SS$-resolved CCN concentration, respectively. The averaged $SS$ for these fogs was estimated to 0.07%±0.02%, slightly higher than previous estimates in urban, suburban, and coast environments, but significantly lower than that derived from aircraft measurements. During the course of fog, temperature reduction played a critical role in supplying sufficient $SS$ for particles activation.

Each fog event was divided into formation, development, maturity, dissipation stages according to visibility variations. Various mechanism dominated the broadening of DSD at different stages, leading to a shift from a unimodal to a trimodal DSD, with peaks observed around 6, 12, and 23 μm. The formation of trimodal DSD was driven by collision-coalescence mechanism during the mature stage of fog, characterized by the $D_{eff}$ exceeding 12 μm. Meanwhile, analysis on the activity of cloud residual particles suggests that apart from small fog droplets, certain un-activated aerosol particles were scavenged by the uptake of larger fog droplets in this stage. For fog events occurring under high $N_a$ background, a notable decrease of temperature during mature stage promoted a secondary activation-dominated process, resulting in the formation of numerous small fog droplets and reducing large droplet size.

The visibility parameterization schemes based on fog microphysical parameters are widely used to estimate fog visibility. The fitting parameters of different $VIS$ parameterization scheme were re-established based on our measuring data. The comparison results indicates that the dual-parameters scheme of $LWC \cdot N_d$ can better describes visibility degradation contributed by fog droplets. However, estimation of fog visibility based on only fog droplet extinction led to substantial deviations when 100 m < $VIS$ ≤ 1000 m. The deviations were notably reduced by incorporating the extinction caused by fog interstitial particles. These findings emphasize the necessity of incorporating both fog droplet and aerosol extinction in fog visibility forecasts, particularly in regions impacted by anthropogenic pollution.

**Data availability.** All data in this paper are available from the authors upon request (liuq@cma.gov.com).

**Author contributions.** Conceptualization: QL and JS. Investigation: QL, XS, JS, YZ, BQ, QM, LH, HX, XH, JL, SL, and AY. Funding acquisition: QL, XS, and HC. Resource: QL, JS, XS, YZ, and XZ. Writing – original draft preparation: QL. All co-authors discussed the results and commented on the manuscript.

**Competing interests.** The authors declare that they have no conflict of interest.

**Acknowledgments.** This work was financially supported by the National Natural Science Foundation of China (grant nos. 42275121, 42030608, 42475121, 42275098), China Meteorological Administration (CXFZ2024J039), and the Chinese Academy of Meteorological Sciences (grant nos. 2023Z012, 2024Z006, 2022KJ002).

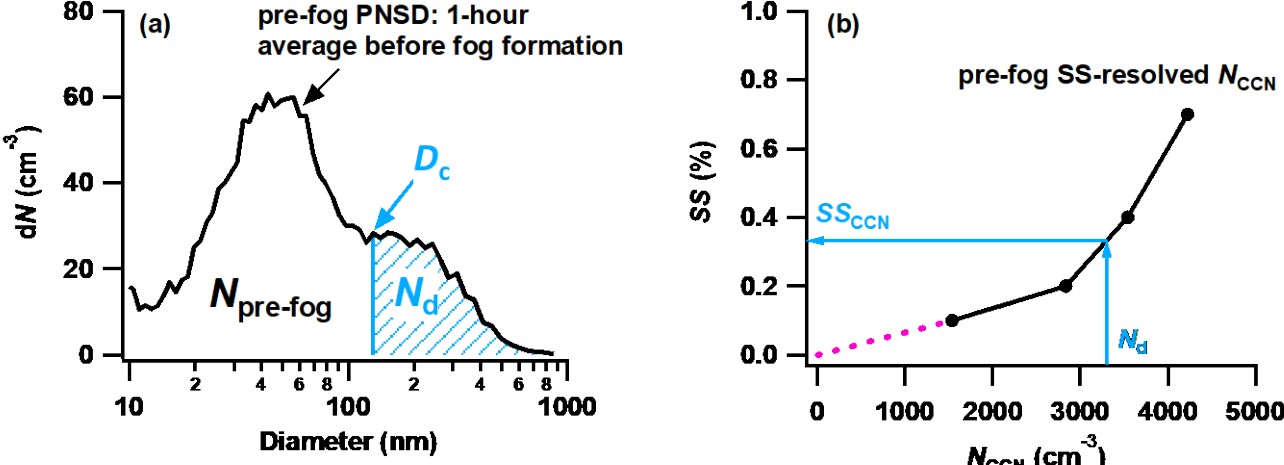

**Fig. 1.** Schematics of two methods for deriving water vapor supersaturation (*SS*) in fog, (a) $SS_{PNSD}$ is derived from the averaged pre-fog particle number size distribution (PNSD) and $N_d$. The blue shaded area represents the integrated $N_a$ from the upper end of the pre-fog PNSD to smaller sizes. The critical activation diameter ($D_c$) is defined as the diameter where the integrated $N_a$ equals $N_d$. (b) $SS_{CCN}$ is derived from the pre-fog SS-resolved $N_{CCN}$ measurements and $N_d$. The magenta dashed line represents linear interpolation from $N_{CCN}$ measurements at two lower SS setpoints.

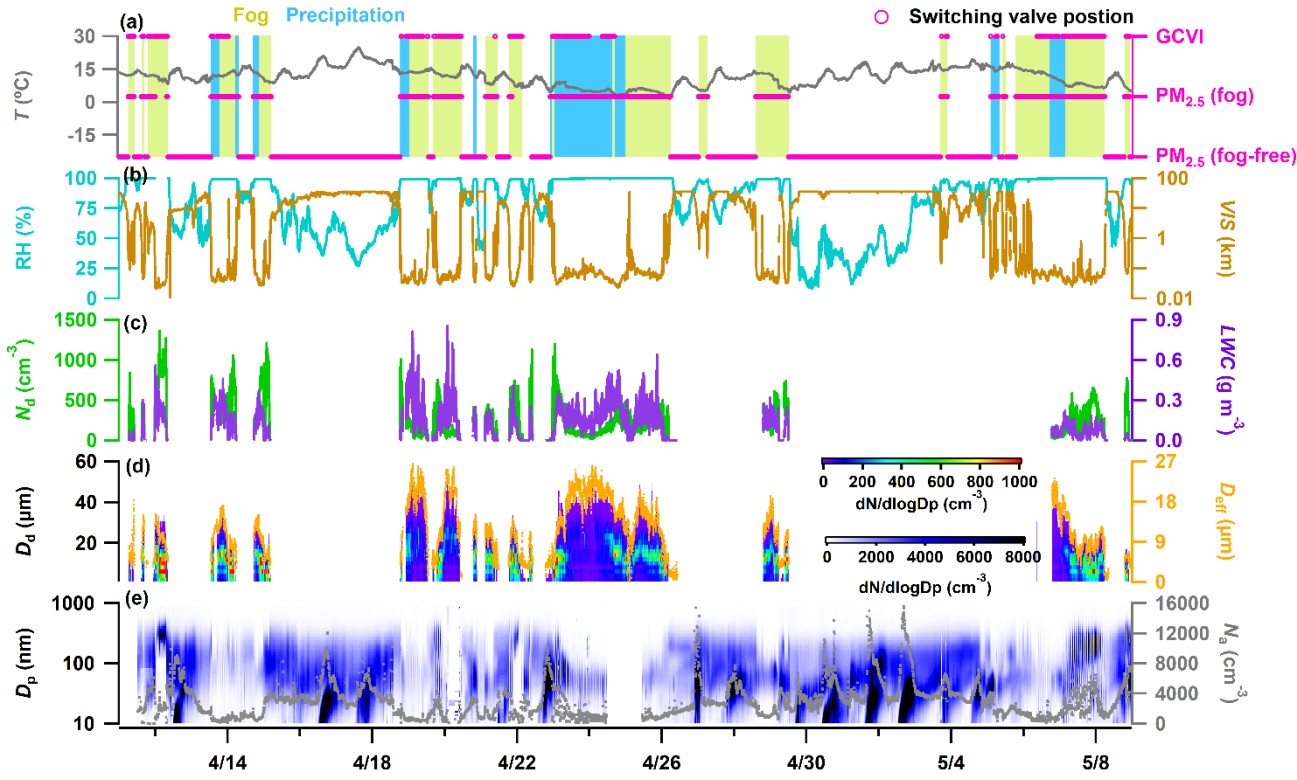

Fig. 2. Time series of (a) temperature ($T$), weather conditions and valve position of the switching inlet system, (b) relative humidity (RH) and visibility (VIS), (c) fog droplet number concentration ($N_d$) and liquid water content (LWC), (d) fog droplets size distribution and effective diameter ($D_{eff}$), and (e) number size distribution and number concentration ($N_a$) of dry aerosol particles, during this campaign. The $D_d$ and $D_p$ in panels (d) and (e) denote the diameters of fog droplets and aerosol particles, respectively.

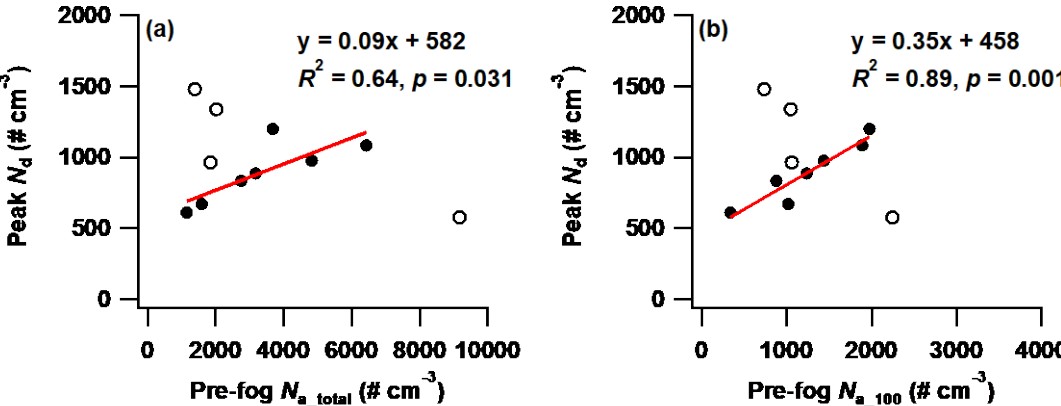

**Fig. 3. Peak $N_d$ value for each fog event versus averaged pre-fog $N_a$ in the last hour before the event, measured by TSMPS within the ranges of (a) total measured sizes (10-850 nm) and (b) sizes larger than 100 nm. Hollow circles represent fog events occurring after precipitation, which are excluded from the correlation analysis.**

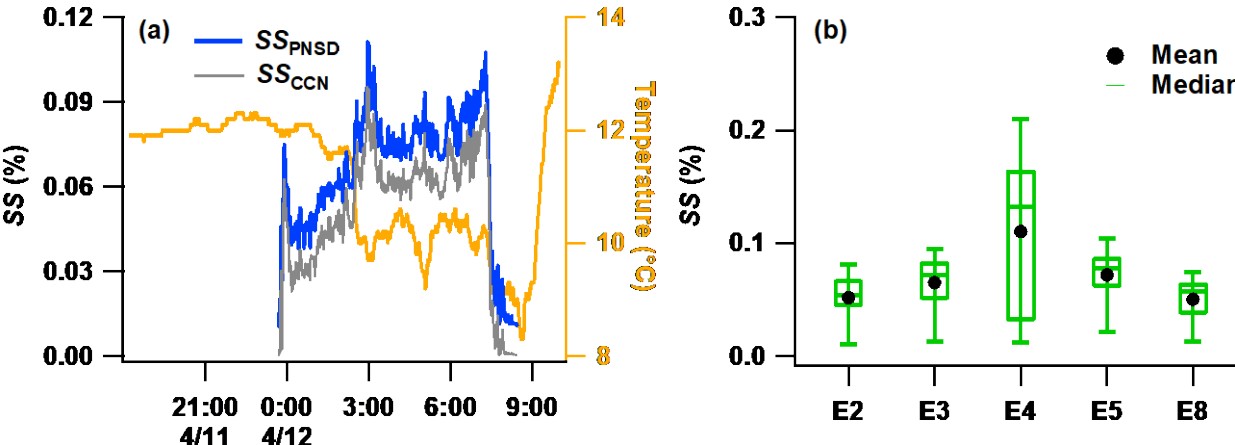

Fig. 4. Estimated water vapor supersaturation ($SS$) in fogs. (a) Temporal variations of $SS_{PNSD}$, $SS_{CCN}$, and temperature during a typical fog event (E3). (b) Statistics of $SS$ for the five available fog events.

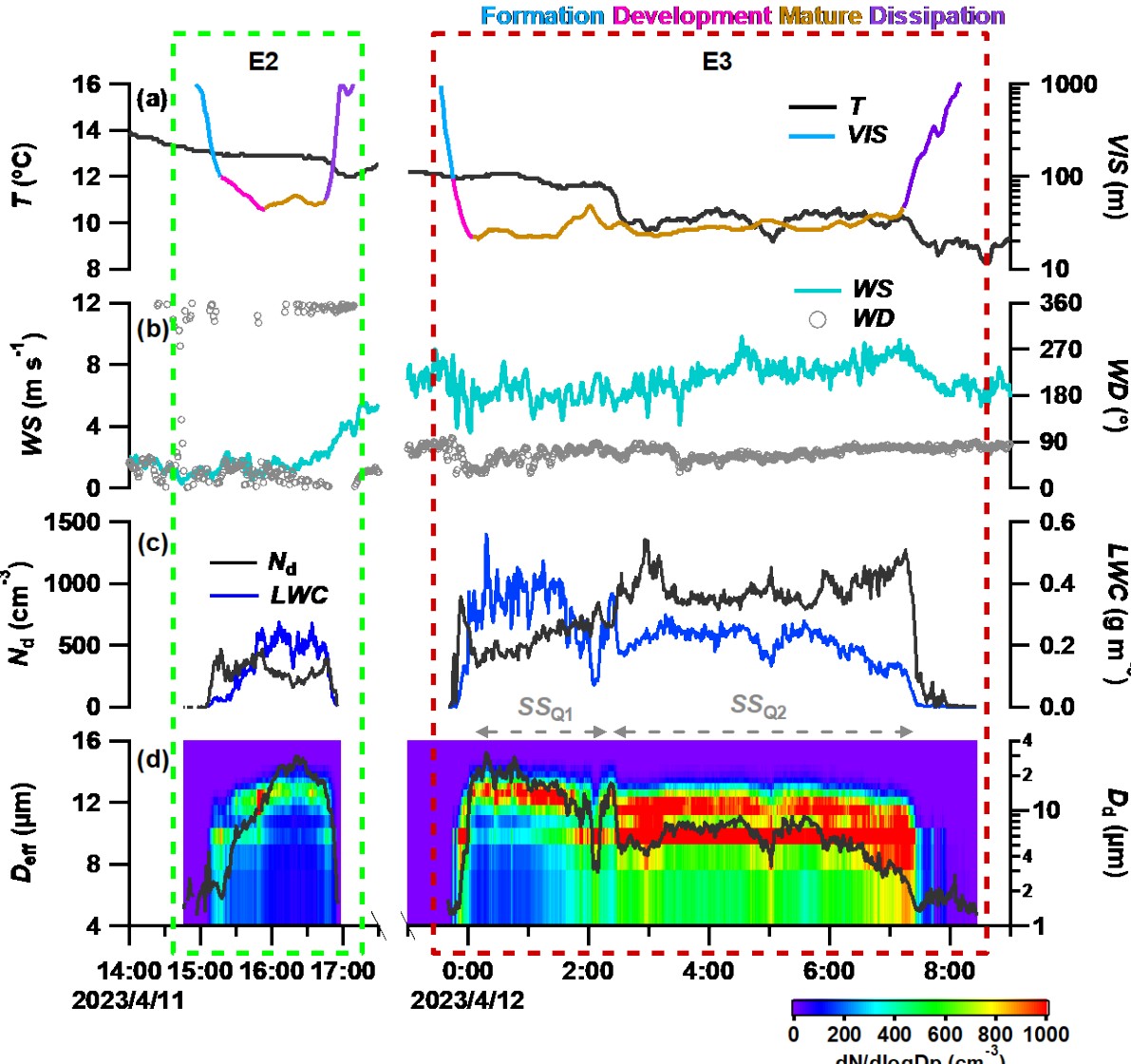

**Fig. 5. Temporal evolution of meteorological parameters and fog microphysical characteristics for two typical fog events, including (a) temperature ($T$) and visibility ($VIS$), (b) wind speed ($WS$) and wind direction ($WD$), (c) fog droplet number concentration ($N_d$) and liquid water content ($LWC$), (d) fog droplets size distribution and effective diameter ($D_{eff}$). E2 represents fog occurring under low pre-fog $N_a$ background, while E3 represents fog occurring under high pre-fog $N_a$ background. The colored lines separate each fog event into four stages based on the evolution of visibility.**

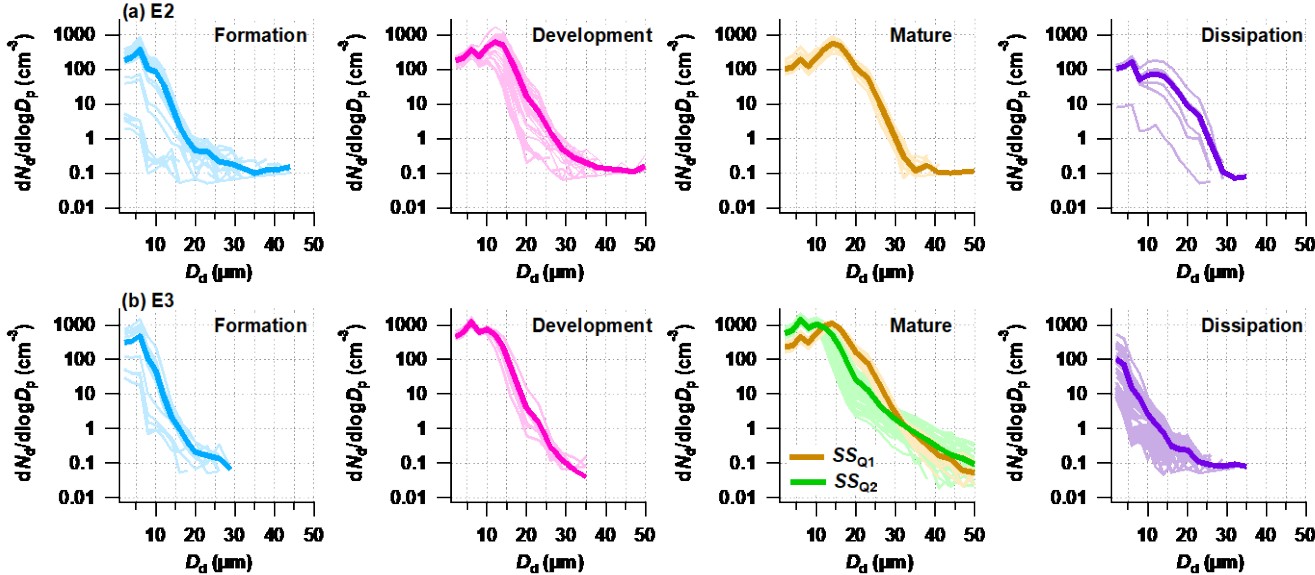

**Fig. 6. Evolutions of fog droplets size distribution (DSD) at various stages during (a) E2 and (b) E3, respectively. Thin lines in each stage represent 1-min averaged DSDs, while the thick line is their average. The $SS_{Q1}$ and $SS_{Q2}$ in panel (b3) represent the first and second quasi-stationary supersaturation states, respectively. $D_d$ denote the diameters of fog droplets**

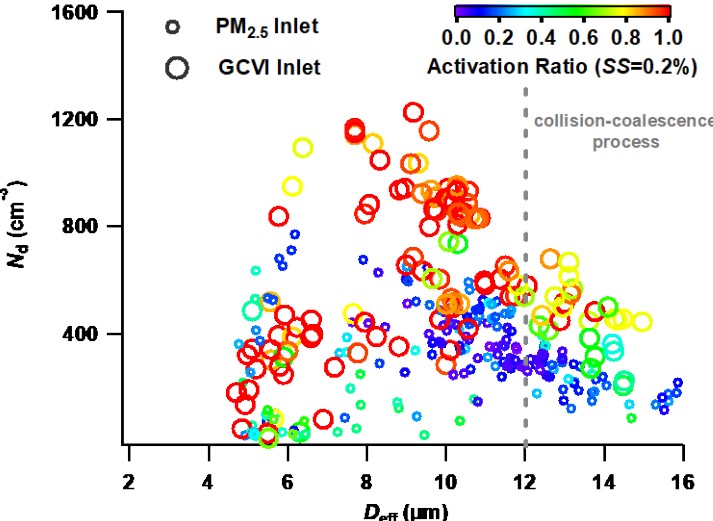

**Fig. 7. Differences in CCN activity between fog residual particles (GCVI inlet) and fog interstitial particles (PM2.5 inlet), and their variations with fog microphysical parameters. The gray dash line indicates significant collision-coalescence processes occurring when $D_{eff}$ exceeds 12 μm.**

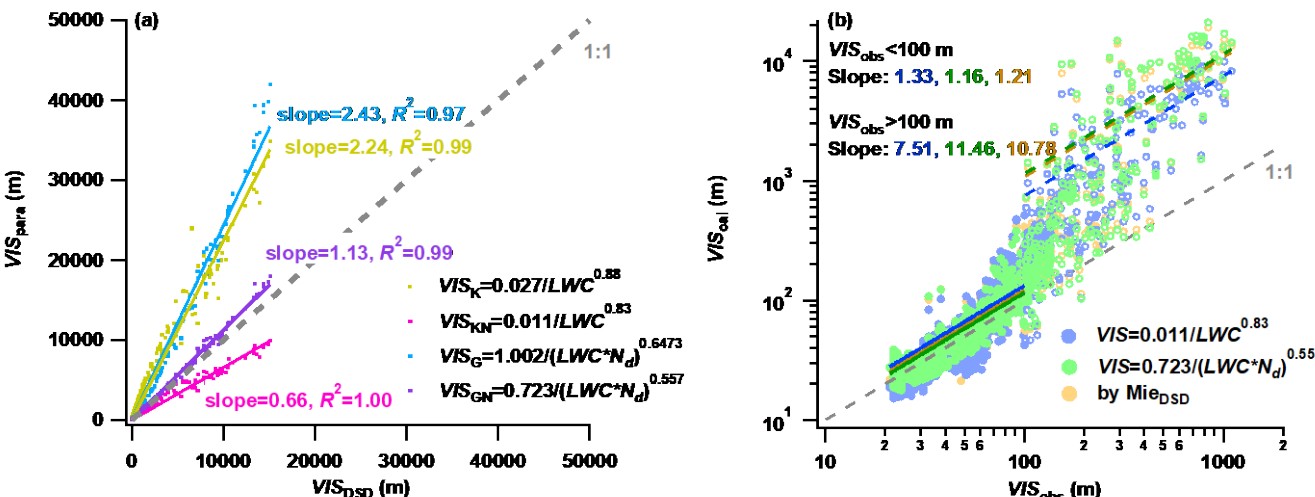

**Fig. 8. Estimation of fog visibility using different calculation methods. (a) Comparison of various visibility parameterization schemes with that derived from droplets size distribution ($VIS_{DSD}$). (b) Relationship between calculated visibility ($VIS_{cal}$) and observed visibility ($VIS_{obs}$). Solid lines represent linear fits of different calculation methods with $VIS_{obs} \leq 100$ m, while dashed lines represent fits for $100$ m $< VIS_{obs} \leq 1000$ m.**

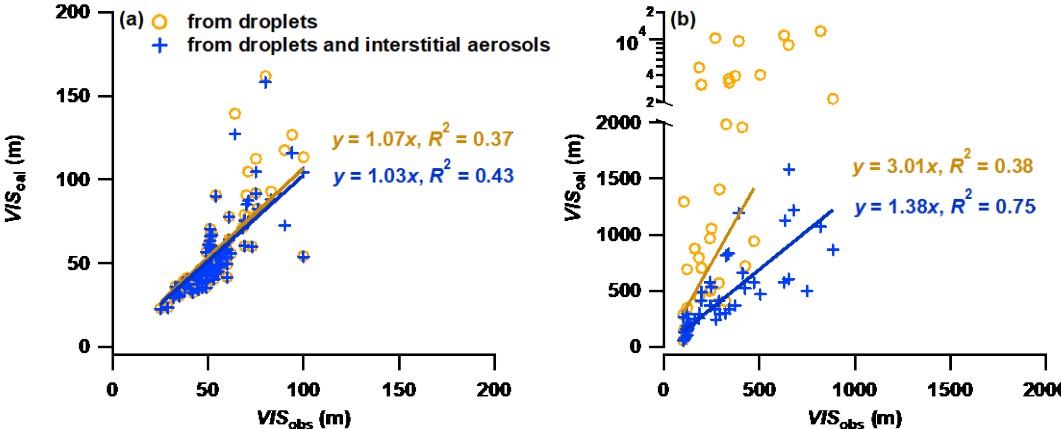

**Fig. 9. Estimating visibility based on only fog droplets and both fog droplets and interstitial particles, respectively. (a) $VIS_{obs}\leq100$ m; (b) 100 m $< VIS_{obs}\leq1000$ m. Note that, values of $VIS_{cal}$ larger than 2000 m have been excluded from the linear fit due to their substantial deviation.**

**Table 1 The median values of measured fog microphysical parameters for each fog event during the campaign.**

| Fog Events | $N_d$ (# cm$^{-3}$) | LWC (g m$^{-3}$) | $D_{eff}$ (μm) |
|---|---|---|---|
| E1 (04/11 06:08 – 10:16) | 146 | 0.009 | 5.9 |
| E2 (04/11 14:55 – 17:00) | 276 | 0.167 | 12.2 |
| E3 (04/11 23:40 – 04/12 08:25) | 834 | 0.216 | 9.9 |
| E4 (04/21 02:50 – 11:15) | 305 | 0.107 | 10.6 |
| E5 (04/21 18:46 – 04/22 03:50) | 469 | 0.116 | 9.7 |
| E6 (04/28 13:30 – 04/29 12:10) | 312 | 0.160 | 11.6 |
| E7 (05/06 18:50 – 05/08 05:55) | 231 | 0.068 | 10.0 |
| E8 (05/08 19:05 – 05/08 22:05) | 504 | 0.025 | 5.5 |
| **Total** | 347 | 0.146 | 10.6 |

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
