# Peer review of "Characterization of fog microphysics and their relationships with visibility at a mountain site in China"

_EGUsphere, 2024_

## Referee Comment (RC1)

Eight fog events are observed and analyzed in this manuscript, with a focus on the characterization of fog microphysics and their relationships with visibility. This is a meaningful study that will likely attract the attention of ACP readers. However, I struggled with the manuscript for the following reasons:

**Major comments**

1.  Analysis of Pre-Fog Aerosols

In Section 3.2, the authors explore the relationship between pre-fog aerosols and fog droplets. Under stable conditions, this relationship is logically sound due to weak wind speed. However, the article reports that wind speed during observation is relatively high (4 to 8 m/s), which suggests that advection plays a significant role in these fog events. The authors also state that "the pre-fog aerosols measured at the observation site may not fully represent the particles that actually activated into fog droplets." This raises the question: Can pre-fog aerosols be reliably replaced by aerosols observed during fog? The rationale behind this needs further explanation. Additionally, how does Section 3.2 lay the foundation for the subsequent content? The logic in Section 3.2 should be clarified.

In Section 3.3, pre-fog aerosols are used in the estimation by the $\kappa$-Köhler equation. How can the authors be certain that the pre-fog aerosols and those that activated into fog droplets share similar physical and chemical properties? For instance, fog event E3 had a long lifetime. Are the changes in aerosol physicochemical properties negligible? Observing supersaturation in fog is challenging, and bias is inevitable. The authors should discuss the sources of errors in this algorithm and provide references to support this approach. Wang et al. (2021) can be referenced.

2.  Mechanism in Fog Event E3

The authors note that "the main wind speeds ranged from 4 to 8 m/s" in lines 157-158, indicating that advection influences the observations. In lines 256-258, they state, "The enhanced supersaturation facilitated the further activation of smaller particles that

were un-activated during the $SS_{Q1}$ stage, resulting in a secondary activation-dominated process during E3." Does this imply that un-activated aerosols from the $SS_{Q1}$ stage remained stationary without being affected by advection? This statement is confusing and potentially misleading.

The authors also mention "excess water vapor" in line 258. However, Figure 4 shows an increase in supersaturation from the $SS_{Q1}$ stage to the $SS_{Q2}$ stage during E3. Does lower supersaturation correspond to excess water vapor during the $SS_{Q1}$ stage? Please clarify this analysis.

In line 261, the authors discuss the "evaporation of liquid water from previously formed large fog droplets." Both large and small droplets are affected by evaporation, but small droplets are more susceptible to dry air because of a larger surface area concentration. The authors only mention large droplets in this context. Moreover, under the influence of advection, even if previous large droplets evaporate, they may not affect current observations. Is this correct? I suggest revising the analysis to clarify the mechanism.

**Minor Comments**

1. There is a formatting issue. When there is no space before a paragraph, a blank line should be inserted between consecutive paragraphs (e.g., a blank line is needed between lines 42 and 43). Alternatively, please refer to the formatting style of articles already published in ACP.

2. In line 37, the article focuses on mountain fog; there is no need to mention maritime fog in the introduction.

3. Distinction Between Clean and Polluted Backgrounds

In lines 159-163, the authors differentiate between clean and polluted backgrounds based on fog microphysical properties. However, the distinction between clean and polluted backgrounds should be based on aerosol concentration, as fog microphysics are also influenced by meteorological conditions. The concentration of cloud

condensation nuclei (CCN) at the same supersaturation level would be more appropriate for this distinction. Numerous studies, such as Figure 2 in Wang et al. (2024), provide CCN concentration data under different background conditions.

4. In Section 2.1, the authors mention that the observation site is far from Hangzhou but claim that the site is generally near the top of the planetary boundary layer (PBL) around midday based on the PBL height of Hangzhou. This is unreliable because the boundary layer height varies by location.

5. The installation of instruments is important for observation results. Could you provide photos of the observation setup in the supplement? This would help readers better understand the instrument installation.

6. In line 145, the threshold involved in the definition of fog requires a reference for support.

7. The information in the figures should be clearly explained. For instance, there is a lack of explanation for Dp in Figure 1; Q1 and Q2 are not explained in the title of Figure 6. Please check other figures.

8. In line 158, there is an "s" at the end of "speeds." Is speed a countable noun?

9. Water Vapor Consumption in Line 218

The hygroscopic growth of aerosols affects the water vapor mixing ratio, but temperature directly influences the saturated water vapor mixing ratio, not water vapor itself. The authors mention only water vapor consumption in line 218. Please reorganize the explanation to clarify the mechanism behind the relatively high supersaturation.

10. Definition of Activation Ratio in Line 243

The authors define the Activation Ratio (AR) as "the CCN number concentration at a supersaturation setting of 0.2% relative to the total particle concentration." Why was 0.2% chosen? Please provide a reference to justify this choice.

11. In line 270, why was 880 nm used in this study? Please provide a reference or explanation.

12. In lines 296-299, the "$\leq$" symbol is not in Times New Roman font.

13. Introduction

In line 68, the authors focus on polluted regions. The criterion for distinguishing between polluted and clean backgrounds is aerosol mass concentration, but the authors do not use this threshold to determine whether the observation site is polluted or clean. Describing the background as having high or low aerosol loading would be more accurate. If the authors wish to continue using the terms "polluted" and "clean," they should provide criteria to support these distinctions.

In lines 67-68, The authors emphasize the impact of interactions between aerosols and fog microphysics on visibility ("their impacts on visibility degradation"). However, only the effect of aerosols on visibility is highlighted. What about the influence of interactions between aerosols and fog on visibility? Additionally, while the effect of aerosols on fog microphysics is analyzed in the manuscript, the effect of fog on aerosols is not addressed (Qian et al., 2023). The interactions between aerosol and fog should be more prominently discussed.

14. There are large uncertainties in the aerosol–cloud interactions (ACIs) (Fan et al., 2016). If the conclusion provides novel insights into ACIs based on the findings related to interactions between aerosols and fog, it could significantly enhance the manuscript's appeal and attract more attention.

**References**

Fan, J., Wang, Y., Rosenfeld, D., and Liu, X.: Review of Aerosol–Cloud Interactions: Mechanisms, Significance, and Challenges, J. Atmos. Sci., 73, 4221-4252, https://doi.org/10.1175/jas-d-16-0037.1, 2016.

Qian, J., Liu, D., Yan, S., Cheng, M., Liao, R., Niu, S., Yan, W., Zha, S., Wang, L., and Chen, X.: Fog scavenging of particulate matters in air pollution events: Observation and simulation in the Yangtze River Delta, China, Sci. Total Environ., 876, 162728, https://doi.org/10.1016/j.scitotenv.2023.162728, 2023.

Wang, Y., Niu, S., Lu, C., Lv, J., Zhang, J., Zhang, H., Zhang, S., Shao, N., Sun, W., Jin, Y., and Song, Q.: Observational study of the physical and chemical

characteristics of the winter radiation fog in the tropical rainforest in Xishuangbanna, China, Sci. China, Ser. D Earth Sci., 64, 1982-1995, https://doi.org/10.1007/s11430-020-9766-4, 2021.

Wang, Y., Li, J., Fang, F., Zhang, P., He, J., Pöhlker, M. L., Henning, S., Tang, C., Jia, H., Wang, Y., Jian, B., Shi, J., and Huang, J.: In-situ observations reveal weak hygroscopicity in the Southern Tibetan Plateau: implications for aerosol activation and indirect effects, npj Clim. Atmos. Sci., 7, https://doi.org/10.1038/s41612-024-00629-x, 2024.

---

## Author Comment (AC1)

**Response to referee' comments on "Characterization of fog microphysics and their relationships with visibility at a mountain site in China"**

**Reviewer 2**

**General comment:**

This manuscript presents an observational study of fog microphysics using measurements collected at a mountain site and tests several visibility estimation parameterizations based on in situ data. The results are clearly presented and could contribute meaningfully to short-term visibility forecasting during fog events. I believe the topic is appropriate for ACP. However, I have the following concerns that should be addressed before considering this work for publication.

[*Response*] We thank the reviewers for their thoughtful and constructive comments that help us improve the manuscript substantially. We have revised the manuscript accordingly. Listed below is our point-to-point response in blue to each comment that was offered by the reviewers. We hope that our revised manuscript will now be suitable for publication in ACP.

**Major**

1. Paper structure

By the end of the Introduction section, you should introduction the structure of the remaining of the manuscript.

Figures 4a-4c are methodology while the panel 4d is a result. You may consider to split this figure and move panels 4a-4c up to the method section.

Section 3.5.1: this section presents previous parameterizations of VIS. Part of the text should be moved to Introduction part and part of it should be moved to methodology. This part can also serve as your motivation of testing the parameterizations using measurements from the mountain site. The results and relevant discussion should remain in this section.

[*Response*] Thanks for pointing these out. We have added the relevant introduction of the structure for the remaining sections as followings:

"In this study, eight fog events are discussed in detail to illustrate the potential impacts of different aerosol concentration background on fog microphysical characteristics. Details on the observation site, instrumentation, sampling inlet system for fog interstitial particles and fog residual particles, and the SS estimation methods are described in the Measurement and methodology section. In the Results and discussions section, we first present general observations during this campaign in Section 3.1 and discuss the relationship between pre-fog aerosols and fog droplets in Section 3.2. Then, the variations of SS values derived by aerosol and fog measurements are presented in Section 3.3. The temporal evolution of fog DSD for two typical fog events is characterized and discussed in Section 3.4. Finally, the contributions of aerosols and droplets to visibility during different stages of fog evolution are presented in Section 3.5. The summaries are provided in the Conclusions and implications section."

For Fig. 4, we adopt your suggestions and move the Fig. 4a-4b to the Methods section (Section 2.2.5). Fig. 4c is the result of derived *SS* during E3 event. There are some introductions and discussions on it, therefore, we have retained the panel 4c in Fig. 4.

According to the suggestion of the referee, we move part of the content of Section 3.5.1 to the Introduction or Methods sections. Please see Lines 65-68 and Section 2.2.6-2.2.7.

2. Introduction:

This section needs more work. For example, there is no mentioning of aerosol extinction in the intro part until the very end. 'aerosol extinction' appears abruptly without any information on how it is related to VIS or microphysics. Second, the motivation of the study presented in this manuscript does not seem clear to me. You listed quite several past studies on fog microphysics and VIS, what are their disadvantages or limitations? What are the values of your work will add to the current understanding or parameterization in terms of VIS forecast? Why this work is necessary given the abundant of work have been done in the past?

[*Response*] Thanks for pointing these out. The calculations of aerosol extinction from particle number size distribution are similar with that of droplets. We added the relevant description in Section 2.2.6.

Regarding the motivation for this study, we acknowledge that numerous studies have explored the relationship between cloud microphysics and visibility. However, the parameterization schemes in those studies were derived from observations in relatively clean areas, where visibility degradation is predominantly caused by fog droplets. These schemes would induce in large uncertainties in visibility calculations in polluted areas, such as the North China Plain (Zhang et al., 2014), where aerosol concentration and extinction contribution can be much higher, especially in light fogs. Additionally, many previous studies have primarily focused either on the effects of haze particles on visibility under subsaturated conditions or on the effects of fog droplets on visibility under supersaturated conditions. However, studies on the contribution of hygroscopic growth of unactivated aerosol particles under supersaturated conditions to visibility are limited. This study conducted simultaneous measurements of aerosol particles and fog droplets to examine their contributions to visibility at different stages of fog evolution. These motivations for the study have been added to the Introduction. Please see Lines 68-74.

3. Incomplete descriptions of the presented figures and lack of discussions:

You seem to only described Figure 1a in Section 3.1, while there are ample information shown in Figures 1b-1e that should be described and discussed.

From my reading, only Figure 4, Figure 5 and Figure 6 are described in detail (while lacking specific reference to the panels in the main text). The rest of the figures deserve more detailed discussions.

[*Response*] Thanks for pointing these out. We add the descriptions and discussions for these figures. For example, the relevant information for Figure 1 has been added as follows:

Lines 201-205: "The visibility variations at this site exhibited distinct characteristics, with values predominantly concentrated in high and low ranges (Fig. 1b), without the gradual increase or decrease typically observed in urban areas (Qiang et al., 2015; Wang

et al., 2015). Moreover, When $RH < 75\%$, the visibility remained above 10 km, whereas it declined below 1 km when $RH > 95\%$. This indicated that low-visibility events at the site were predominantly driven by fog processes during the observation period."

Lines 210-213: "The variations of $N_d$ and $LWC$ showed a consistent trend during fog formation and dissipation stages. However, after fog formation, the trends of the two variables may diverge (Fig. 1c), which is closely related to the variations in $D_{eff}$ (Fig. 1d). The relationship between $N_d$ and $LWC$ during the 8 available fog events is presented in Fig. S4 to further illustrate their correlation."

Lines 219-224: "Although there were few anthropogenic sources near the site, the observed aerosol concentrations varied dramatically. As shown in Fig. 1e, the $N_a$ ranged from 230 to 15620 cm$^{-3}$, with a median of 2750 cm$^{-3}$. Episodes with $N_a$ exceeding 8000 cm$^{-3}$ were typically associated with a pronounced increase in the concentration of small particles within a range of 10-100 nm (Fig. 1e), which were likely driven by new particle formation (Shen et al., 2022). In the subsequent discussion, the pre-fog aerosol concentration below and above this median were defined as low and high aerosol loading backgrounds, respectively."

4. Grammar errors

I found many grammar errors in the abstract. I tried to capture some of them in my minor comments, but they are by no means a complete list. I did not list any grammar errors in the main text. The authors should do a thorough proof reading before resubmitting.

[*Response*] Thanks for pointing these out. We have carefully checked the entire manuscript and corrected the grammar errors.

**Minor**

Line 17: Clarify whether the elevation of 1483 m is above ground level or mean sea level.

[*Response*] Above mean sea level

Line 18: Consider rephrasing to, "In this study, eight fog events were investigated during the campaign, ..."

[*Response*] Revised.

Line 23: Add "and collision-coalescence mechanisms."

[*Response*] Revised.

Line 24: Rephrase as, "Peaks were observed at around ..."

[*Response*] Suggestion adopted.

Citation Format: When citing a reference at the beginning of a sentence (e.g., "Song et al. (2019) found that ..."), you do not need to cite it again in parentheses at the end of the sentence.

[*Response*] Revised. We remove the repetitive citation at the end of the sentence.

Line 164-167, and Equation 1-3: The linear relationship between LWC and Nd within a specific Deff bin seems expected based on equations 1–3. Since D_eff is the ratio of the third to second moments, it can be treated as particle size, meaning that LWC should increase with higher Nd. Can you clarify or further discuss this?

[*Response*] As pointed out by the referee, these three parameters in Fig. 2 are derived from the observed droplets size distribution and Equations 1-3. Their relationship should be the outcome of Equations 1-3. The purpose of this figure is to highlight that $N_d$ generally decreases as $D_{eff}$ increases within a given range of *LWC* values. This negative correlation between them is ubiquitous in fog, as the presence of more droplets competes for available water vapor, thereby inhibiting their growth (Li et al., 2017). This serves as a foundation to the subsequent discussion on the evolution of fog droplets size distribution. We have moved it to the supplementary material (Fig. S4).

Lines 183-184: Could the difference in findings between this study and the previous one be due to the different elevations of the measurement sites?

[*Response*] The slope of the linear relationship between peak $N_d$ and pre-fog $N_a$ can represent the bulk activation rate of aerosol particles, which is depended on aerosol physicochemical properties and ambient water vapor supersaturation (*SS*) conditions. As the discussion in Section 3.3, compared with previous studies, the estimated *SS* in various observation environments seems to be positively correlated with altitude. This can be partly attributed to the lower aerosol number concentration and temperature at high altitudes (Liu et al., 2020), which reduce excess water vapor consumption in clouds and fog, as well as the equilibrium vapor pressure (Baccarini et al., 2020; Shen

et al., 2018), thereby promoting supersaturation. Therefore, the difference in the slope between this study and the previous one can be attributed to both different aerosol properties and $SS$ conditions in the studies. We add the relevant discussion in the revised manuscript. Please see Lines 269-272.

Line 198: When you mention the second approach, do you mean $N_d$ is equivalent to $N_{CCN}$? Please clarify.

[*Response*] Thanks for pointing this out. In the second approach, the $N_d$ in the fog is considered to be consistent with the activated CCN number concentration ($N_{CCN}$). Therefore, the $SS_{CCN}$ was determined as the $N_d$ is equivalent to $N_{CCN}$ by using linear interpolation of the pre-fog $SS$-resolved $N_{CCN}$ measurements. We have clarified it in the revised manuscript, please see Lines 169-170.

Lines 213-215: Are the studies you compare your results to all focused on fog events, or do any deal with clouds, such as the Gong et al. paper?

[*Response*] We have confirmed it again. The studies we used to compare the $SS$ in different environments all focused on fog events except Gong et al. (2023). The $SS$ in Gong et al. (2023) was derived from aircraft measurements of clouds. We have rewritten the sentence to make it clear. Please see Lines 265-268.

Line 225: The VIS during the development stage of the 04/12 event does not appear to decrease at a slower rate compared to the formation stage. Did you apply specific thresholds for the rate of change of VIS to define these stages? If so, please justify how these thresholds were determined.

[*Response*] According to previous studies (Mazoyer et al., 2022; Niu et al., 2010; Pilie et al., 1975), the stages during fog event were mainly determined by the thresholds of $VIS$ value. As it shown in Fig. R1, there are 12 data points with the $VIS$ decreasing from 967 m to 100 m in the formation stage, but 20 data points with the $VIS$ decreasing from 95 m to 23 m in the development stage. Although there is no specific threshold for the rate of change of $VIS$, the decrease rate of $VIS$ in the development stage (~4 m min$^{-1}$) was much slower than that in the formation stage (~72 m min$^{-1}$) during the 04/12 event.

[Figure]

Fig. R1 Temporal evolution of meteorological parameters and fog microphysical characteristics for two typical fog events, including (a) temperature ($T$) and visibility ($VIS$), (b) fog droplet number concentration ($N_d$) and liquid water content ($LWC$), (c) fog droplets size distribution and effective diameter ($D_{eff}$). E2 represents fog occurring under low pre-fog Na background, while E3 represents fog occurring under high pre-fog $N_a$ background. The colored lines separate each fog event into four stages based on the evolution of visibility.

Lines 230-244: It would be helpful to include specific figure and panel numbers after each discussion sentence, particularly when referring to DSD descriptions, to make it easier for readers to follow along. This is especially important in the case of the E3 event.

[*Response*] Thanks for pointing this out. We have added the specific figure and panel numbers after the corresponding discussions in the revised manuscript.

References

Baccarini, A., Karlsson, L., Dommen, J., Duplessis, P., Vüllers, J., Brooks, I. M., Saiz-Lopez, A., Salter, M., Tjernström, M., Baltensperger, U., Zieger, P., and Schmale, J.: Frequent new particle formation over the high Arctic pack ice by enhanced iodine emissions, Nature communications, 11, 4924, 10.1038/s41467-020-18551-0, 2020.

Gong, X., Wang, Y., Xie, H., Zhang, J., Lu, Z., Wood, R., Stratmann, F., Wex, H., Liu, X., and Wang, J.: Maximum Supersaturation in the Marine Boundary Layer Clouds Over the North Atlantic, AGU Advances, 4, e2022AV000855, https://doi.org/10.1029/2022AV000855, 2023.

Li, S., Joseph, E., Min, Q., and Yin, B.: Multi-year ground-based observations of aerosol-cloud interactions in the Mid-Atlantic of the United States, Journal of Quantitative Spectroscopy and Radiative Transfer, 188, 192-199, https://doi.org/10.1016/j.jqsrt.2016.02.004, 2017.

Liu, Q., Liu, D., Gao, Q., Tian, P., Wang, F., Zhao, D., Bi, K., Wu, Y., Ding, S., Hu, K., Zhang, J., Ding, D., and Zhao, C.: Vertical characteristics of aerosol hygroscopicity and impacts on optical properties over the North China Plain during winter, Atmos. Chem. Phys., 20, 3931-3944, 10.5194/acp-20-3931-2020, 2020.

Mazoyer, M., Burnet, F., and Denjean, C.: Experimental study on the evolution of droplet size distribution during the fog life cycle, Atmos. Chem. Phys., 22, 11305-11321, 10.5194/acp-22-11305-2022, 2022.

Niu, S., Lu, C., Liu, Y., Zhao, L., Lü, J., and Yang, J.: Analysis of the microphysical structure of heavy fog using a droplet spectrometer: A case study, Advances in Atmospheric Sciences, 27, 1259-1275, 10.1007/s00376-010-8192-6, 2010.

Pilie, R. J., Mack, E. J., Kocmond, W. C., Rogers, C. W. C., and Eadie, W.: The Life Cycle of Valley Fog. Part I: Micrometeorological Characteristics, Journal of Applied Meteorology, 14, 347-363, 1975.

Qiang, Z., Jiannong, Q., Xuexi, T., Xia, L., Quan, L., Yang, G., and Delong, Z.: Effects of meteorology and secondary particle formation on visibility during heavy haze events in Beijing, China, Science of the Total Environment, 502, 578-584, 2015.

Shen, C., Zhao, C., Ma, N., Tao, J., Zhao, G., Yu, Y., and Kuang, Y.: Method to Estimate Water Vapor Supersaturation in the Ambient Activation Process Using Aerosol and Droplet Measurement Data, Journal of Geophysical Research: Atmospheres, 123, 10,606-610,619, https://doi.org/10.1029/2018JD028315, 2018.

Shen, X., Sun, J., Ma, Q., Zhang, Y., Zhong, J., Yue, Y., Xia, C., Hu, X., Zhang, S., and Zhang, X.: Long-term trend of new particle formation events in the Yangtze River Delta, China and its influencing factors: 7-year dataset analysis, Science of The Total Environment, 807, 150783, https://doi.org/10.1016/j.scitotenv.2021.150783, 2022.

Wang, Y. H., Liu, Z. R., Zhang, J. K., Hu, B., Ji, D. S., Yu, Y. C., and Wang, Y. S.: Aerosol physicochemical properties and implications for visibility during an intense haze episode during winter in Beijing, Atmospheric Chemistry and Physics, 15, 3205-3215, 10.5194/acp-15-3205-2015, 2015.

Zhang, J., Xue, H., Deng, Z., Ma, N., Zhao, C., and Zhang, Q.: A comparison of the parameterization schemes of fog visibility using the in-situ measurements in the North China Plain, Atmospheric Environment, 92, 44-50, https://doi.org/10.1016/j.atmosenv.2014.03.068, 2014.

---

## Author Comment (AC2)

**Response to referee' comments on "Characterization of fog microphysics and their relationships with visibility at a mountain site in China"**

**Reviewer 1**

**General comment:**

Eight fog events are observed and analyzed in this manuscript, with a focus on the characterization of fog microphysics and their relationships with visibility. This is a meaningful study that will likely attract the attention of ACP readers. However, I struggled with the manuscript for the following reasons:

[*Response*] We thank the reviewers for their thoughtful and constructive comments that help us improve the manuscript substantially. We have revised the manuscript accordingly. Listed below is our point-to-point response in blue to each comment that was offered by the reviewers. We hope that our revised manuscript will now be suitable for publication in ACP.

**Major comments**

1. Analysis of Pre-Fog Aerosols

In Section 3.2, the authors explore the relationship between pre-fog aerosols and fog droplets. Under stable conditions, this relationship is logically sound due to weak wind speed. However, the article reports that wind speed during observation is relatively high (4 to 8 m/s), which suggests that advection plays a significant role in these fog events. The authors also state that "the pre-fog aerosols measured at the observation site may not fully represent the particles that actually activated into fog droplets." This raises the question: Can pre-fog aerosols be reliably replaced by aerosols observed during fog? The rationale behind this needs further explanation. Additionally, how does Section 3.2 lay the foundation for the subsequent content? The logic in Section 3.2 should be clarified.

In Section 3.3, pre-fog aerosols are used in the estimation by the $\kappa$-Köhler equation. How can the authors be certain that the pre-fog aerosols and those that activated into

fog droplets share similar physical and chemical properties? For instance, fog event E3 had a long lifetime. Are the changes in aerosol physicochemical properties negligible? Observing supersaturation in fog is challenging, and bias is inevitable. The authors should discuss the sources of errors in this algorithm and provide references to support this approach. Wang et al. (2021) can be referenced.

[*Response*] Thanks for pointing this out. Although there is a temporal difference between the observation of pre-fog aerosols and the subsequent fog process at a fixed site, the measured pre-fog aerosol particles may not fully represent the particles that actually activated into fog droplets. However, due to the high altitude of this mountain site, it is located above the top of the boundary layer for most of the day (Sun et al., 2018). At this height, the aerosol concentration and properties are relatively homogeneous within a large spatial range. Although the observed fog droplets were partly formed elsewhere and advected to the site, especially in high wind speed conditions, the aerosol particles at the site are regionally representative, resulting in a good correlation between the pre-fog aerosol and the peak $N_d$ discussed in Section 3.2. Conversely, the good correlation between them also indicated the observations at this site were representative of a relatively large spatial scale. This provides a rational basis for estimating water vapor supersaturation by using the pre-fog aerosol size distribution in Section 3.3. We add these descriptions in the revised manuscript. Please see Lines 231-238. Additionally, we also consider add a sample inlet of total suspended particles in future experiments, which can obtain the information of both aerosol particles and fog droplets. This can help us gain a more comprehensive understanding of the properties of fog residual particles and fog interstitial particles.

As pointed out by the referee, the *SS* estimation algorithm in Section 3.3 considered only adiabatic processes such as activation and condensation, and ignores non-adiabatic processes such as collision-coalescence (Wang et al., 2021). If the reduction of $N_d$ caused by the collision-coalescence process is considered, the actual effective *SS* should be greater than the calculated value. We have added the sources of errors in this algorithm and provide relevant references. Please see Lines 256-259.

2. Mechanism in Fog Event E3

The authors note that "the main wind speeds ranged from 4 to 8 m/s" in lines 157-158, indicating that advection influences the observations. In lines 256-258, they state, "The enhanced supersaturation facilitated the further activation of smaller particles that

were un-activated during the $SS_{Q1}$ stage, resulting in a secondary activation-dominated process during E3." Does this imply that un-activated aerosols from the $SS_{Q1}$ stage remained stationary without being affected by advection? This statement is confusing and potentially misleading.

The authors also mention "excess water vapor" in line 258. However, Figure 4 shows an increase in supersaturation from the $SS_{Q1}$ stage to the $SS_{Q2}$ stage during E3. Does lower supersaturation correspond to excess water vapor during the $SS_{Q1}$ stage? Please clarify this analysis.

[*Response*] Thanks for pointing this out. In-situ observations at a fixed site face significant challenges in continuously measuring the evolution of aerosols and fog droplets within a specific air mass. Here, we assume that at a certain height within the fog, the aerosols and fog droplets exhibit similar microphysical characteristics and undergo similar variations. Therefore, during a fog process, measurements at different time points at this site can, to some extent, reflect the evolution of the microphysical characteristics of aerosols and cloud droplets at that height. We add this assumption in the revised manuscript to clarify it. Please see Lines 281-285.

The excess water vapor mentioned in Line 258 is the difference between the partial pressure of vapor and the equilibrium value. When the production and depletion of excess water vapor in the early mature stage were in approximate balance, the first quasi-stationary supersaturation ($SS_{Q1}$) was reached. As the temperature decreased after the $SS_{Q1}$ state, the temperature-dependent equilibrium vapor pressure decreased faster than the partial pressure of vapor, leading to increases both in excess water vapor pressure and supersaturation during the $SS_{Q2}$ stage. We revised the description to further clarify that mechanism as follows:

Lines 296-298: "This indicated that the excess water vapor, defined as the difference of the ambient water vapor pressure and the equilibrium value, was produced and consumed in approximate balance, thus reaching a quasi-stationary supersaturation state."

Lines 317-322: "However, after reaching and maintaining a quasi-equilibrium supersaturation state ($SS_{Q1}$) in the early mature stage, a notable decrease in temperature occurred (Fig. 5a). This decrease caused an increase in both excess water vapor pressure and supersaturation, as the temperature-dependent equilibrium vapor pressure dropped faster than the ambient partial vapor pressure. Consequently, a new quasi-equilibrium

supersaturation state ($SS_{Q2}$) was established, exhibiting distinct fog microphysical characteristics (Fig. 6b)"

3. In line 261, the authors discuss the "evaporation of liquid water from previously formed large fog droplets." Both large and small droplets are affected by evaporation, but small droplets are more susceptible to dry air because of a larger surface area concentration. The authors only mention large droplets in this context. Moreover, under the influence of advection, even if previous large droplets evaporate, they may not affect current observations. Is this correct? I suggest revising the analysis to clarify the mechanism.

[*Response*] As pointed out by the reviewer, both large and small droplets are affected by evaporation. The discussion here aims to explain the reduction in effective droplet radius. To avoid ambiguity, this is revised as below:
"During this secondary activation process, a greater number of small droplets formed and competed for the limited water vapor, which led to a decrease in the $D_{eff}$ (Fig. 6b)."

**Minor Comments**

1. There is a formatting issue. When there is no space before a paragraph, a blank line should be inserted between consecutive paragraphs (e.g., a blank line is needed between lines 42 and 43). Alternatively, please refer to the formatting style of articles already published in ACP.

[*Response*] Thanks for the reviewer's suggestion. We have formatted the revised manuscript according to published articles in ACP.

2. In line 37, the article focuses on mountain fog; there is no need to mention maritime fog in the introduction.

[*Response*] Thanks for the reviewer's suggestion. We have removed the information of maritime fog from the Introduction in the revised manuscript.

3. Distinction Between Clean and Polluted Backgrounds.
In lines 159-163, the authors differentiate between clean and polluted backgrounds based on fog microphysical properties. However, the distinction between clean and polluted backgrounds should be based on aerosol concentration, as fog microphysics are also influenced by meteorological conditions. The concentration of cloud

condensation nuclei (CCN) at the same supersaturation level would be more appropriate for this distinction. Numerous studies, such as Figure 2 in Wang et al. (2024), provide CCN concentration data under different background conditions.

[*Response*] Thanks for the reviewer's suggestion. In the revised manuscript, the aerosol concentrations have been used to differentiate between low and high number concentrations of aerosol backgrounds. Relevant information has been added in Lines 219-224 as below:

"Although there were few anthropogenic sources near the site, the observed aerosol concentrations varied dramatically. As shown in Fig. 1e, the $N_a$ ranged from 230 to 15620 cm$^{-3}$, with a median of 2750 cm$^{-3}$. Episodes with $N_a$ exceeding 8000 cm$^{-3}$ were typically associated with a pronounced increase in aerosol number concentration within the size range of 100-100 nm (Fig. 1e), which were likely driven by new particle formation (Shen et al., 2022). In the subsequent discussion, the pre-fog aerosol concentration below and above this median were defined as low and high number concentrations of aerosol backgrounds, respectively."

4. In Section 2.1, the authors mention that the observation site is far from Hangzhou but claim that the site is generally near the top of the planetary boundary layer (PBL) around midday based on the PBL height of Hangzhou. This is unreliable because the boundary layer height varies by location.

[*Response*] Thanks for pointing this out. Due to the lack of measurement of the PBL height on this site, we have removed the relevant description from the revised manuscript.

5. The installation of instruments is important for observation results. Could you provide photos of the observation setup in the supplement? This would help readers better understand the instrument installation.

[*Response*] As the reviewer suggested, we have combined the photos and the schematic of instrument setup together as Fig. S2 in the supplement, also shown as below:

[Figure]

**Fig. S2. Schematic of the experimental setup at the Daming Mountain site. An automatic three-way switching system was placed between the sample inlets and instruments. Meteorological parameters and fog droplets were simultaneously measured on the roof of the observation container. The bypass pump only operated when the three-way valve connected to the PM$_{2.5}$ inlet. Its flow rate was controlled at 4.5 L min$^{-1}$ via a mass flow controller, ensuring the total sample flow reached the 16.7 L min$^{-1}$ required by the PM$_{2.5}$ cyclone inlet.**

6. In line 145, the threshold involved in the definition of fog requires a reference for support.

[*Response*] Suggestion adopted. We have added the relevant references in Lines 154-155 as follows:

Deng, Z., Zhao, C., Zhang, Q., Huang, M., and Ma, X.: Statistical analysis of microphysical properties and the parameterization of effective radius of warm clouds in Beijing area, Atmospheric Research, 93, 888-896, https://doi.org/10.1016/j.atmosres.2009.04.011, 2009.

Lu, C., Niu, S., Liu, Y., and Vogelmann, A. M.: Empirical relationship between entrainment rate and microphysics in cumulus clouds, 40, 2333-2338, https://doi.org/10.1002/grl.50445, 2013.

World Meteorological Organization: International Cloud Atlas - Manual on the Observation of Clouds and Other Meteors [WWW Document]. WMO-No. 407. URL https://cloudatlas.wmo.int/fog-compared-with-mist.html, 2017.

7. The information in the figures should be clearly explained. For instance, there is a lack of explanation for Dp in Figure 1; Q1 and Q2 are not explained in the title of Figure 6. Please check other figures.

[*Response*] Suggestion adopted. Here, $D_p$ in Figure 1 represents the diameter of droplet or particle. To avoid any confusion between them, we use $D_d$ and $D_p$ to denote the diameters of fog droplets and aerosol particles, respectively. Explanations for $SS_{Q1}$ and $SS_{Q1}$ have been added to the revised figure caption. We have also checked others figures thoroughly.

8. In line 158, there is an "s" at the end of "speeds." Is speed a countable noun?

[*Response*] Revised.

9. Water Vapor Consumption in Line 218

The hygroscopic growth of aerosols affects the water vapor mixing ratio, but temperature directly influences the saturated water vapor mixing ratio, not water vapor itself. The authors mention only water vapor consumption in line 218. Please reorganize the explanation to clarify the mechanism behind the relatively high supersaturation.

[*Response*] Thanks for the reviewer's suggestion. In the revised manuscript, we have revised the interpretation of the positive correlation between estimated *SS* and altitudes as below:

"This can be partly attributed to the lower aerosol number concentration and temperature at high altitudes (Liu et al., 2020b), which reduce excess water vapor consumption in clouds and fog, as well as the equilibrium vapor pressure (Baccarini et al., 2020; Shen et al., 2018), thereby promoting supersaturation."

10. Definition of Activation Ratio in Line 243

The authors define the Activation Ratio (AR) as "the CCN number concentration at a supersaturation setting of 0.2% relative to the total particle concentration." Why was 0.2% chosen? Please provide a reference to justify this choice.

[*Response*] Thanks for pointing this out. To avoid excessive *SS* variation in the CCNc column, the four *SS* setpoints were sequentially scanned from low to high and then back from high to low. Consequently, the number of data points for the intermediate *SS* values is twice that of the endpoint *SS* values. Meanwhile, CCNs with weaker activation are more likely to remain un-activated under low *SS* conditions. Based on the above considerations, the case of *SS* = 0.2% was selected in Fig. 6 to discuss the relationship between them. We have added the results for other *SS* setpoints to the supplement information (Fig. S10), and it can be seen that they present a phenomenon that is basically consistent with the results discussed for *SS* 0.2%. The relevant descriptions had added in the revised manuscript. Please see Lines 303-308.

[Figure]

Fig. S10. Differences in CCN activity between fog residual particles (GCVI inlet) and fog interstitial particles (PM$_{2.5}$ inlet), and their variations with fog microphysical parameters: (a) *SS*=0.1%, (b) *SS*=0.4%, and (c) *SS*=0.7%. The gray dash line indicates significant collision-coalescence processes occurring when *D*$_{eff}$ exceeds 12 μm.

11. In line 270, why was 880 nm used in this study? Please provide a reference or explanation.

[*Response*] The wavelength used in the visibility meter is 880 nm. In order to make the *VIS* derived from the Mie theory is comparable with the *VIS* measured by the visibility meter, the same wavelength was used in the *VIS* calculation. We clarified it in the revised manuscript. Please see Lines 180-183.

12. In lines 296-299, the "≤" symbol is not in Times New Roman font.
[*Response*] Revised.

13. Introduction
In line 68, the authors focus on polluted regions. The criterion for distinguishing between polluted and clean backgrounds is aerosol mass concentration, but the authors do not use this threshold to determine whether the observation site is polluted or clean. Describing the background as having high or low aerosol loading would be more accurate. If the authors wish to continue using the terms "polluted" and "clean," they should provide criteria to support these distinctions.

In lines 67-68, The authors emphasize the impact of interactions between aerosols and fog microphysics on visibility ("their impacts on visibility degradation"). However, only the effect of aerosols on visibility is highlighted. What about the influence of interactions between aerosols and fog on visibility? Additionally, while the effect of aerosols on fog microphysics is analyzed in the manuscript, the effect of fog on aerosols is not addressed (Qian et al., 2023). The interactions between aerosol and fog should be more prominently discussed.

[*Response*] Thanks for your suggestion. The term "polluted region" here referred to the megacity cluster of the YRD region mentioned later in this sentence. The paper did not discuss clean or polluted weather conditions. In Section 3.4, the terms "low aerosol concentration condition" and "high aerosol concentration condition" are used, but their definitions were not provided. Following the referee's suggestion, we have added descriptions for the classification criteria in Lines 219-224.

For the interactions between aerosol and fog on visibility, we have discussed the effects of aerosol concentration on $N_d$ and evolution of fog droplets size distribution. These fog microphysical parameters significantly influence visibility, as discussed in Section 3.5. Additionally, we acknowledge that the effect of fog on aerosols is crucial for understanding the interactions between aerosols and fog. After participating the fog process, the chemical composition, mixing state, and morphology of aerosol particles

would be changed (Schroder et al., 2015; Roth et al., 2016; Qian et al., 2023). At downstream of the GCVI inlet, the TSMPS, AMS and SP2 were also installed to measure physicochemical properties of fog residual particles. The results of these measurements will be used to analyzed the effects of fog on aerosol particles in a subsequent paper.

14. There are large uncertainties in the aerosol–cloud interactions (ACIs) (Fan et al., 2016). If the conclusion provides novel insights into ACIs based on the findings related to interactions between aerosols and fog, it could significantly enhance the manuscript's appeal and attract more attention.

[*Response*] Thanks for your suggestion. In the conclusion, we described the influence of pre-existing aerosol levels on the peak $N_d$ of each fog event and highlighted a secondary activation process that occurred during fog evolution. This process led to the formation of numerous small fog droplets, thus reducing the effective diameter. We acknowledge the effects of fog droplets on aerosol particles are also important for better understanding the interactions between aerosols and fog. Elaborate analysis for these measurements is prepared for a subsequent paper.

**References**

Fan, J., Wang, Y., Rosenfeld, D., and Liu, X.: Review of Aerosol–Cloud Interactions: Mechanisms, Significance, and Challenges, J. Atmos. Sci., 73, 4221-4252, https://doi.org/10.1175/jas-d-16-0037.1, 2016.

Roth, A., Schneider, J., Klimach, T., Mertes, S., van Pinxteren, D., Herrmann, H., and Borrmann, S.: Aerosol properties, source identification, and cloud processing in orographic clouds measured by single particle mass spectrometry on a central European mountain site during HCCT-2010, Atmos. Chem. Phys., 16, 505-524, 10.5194/acp-16-505-2016, 2016.

Qian, J., Liu, D., Yan, S., Cheng, M., Liao, R., Niu, S., Yan, W., Zha, S., Wang, L., and Chen, X.: Fog scavenging of particulate matters in air pollution events: Observation and simulation in the Yangtze River Delta, China, Sci. Total Environ., 876, 162728, https://doi.org/10.1016/j.scitotenv.2023.162728, 2023.

Schroder, J. C., Hanna, S. J., Modini, R. L., Corrigan, A. L., Kreidenwies, S. M., Macdonald, A. M., Noone, K. J., Russell, L. M., Leaitch, W. R., and Bertram, A. K.: Size-resolved observations of refractory black carbon particles in cloud droplets at a marine boundary layer site, Atmos. Chem. Phys., 15, 1367-1383, 10.5194/acp-15-1367-2015, 2015.

Shen, X., Sun, J., Ma, Q., Zhang, Y., Zhong, J., Yue, Y., Xia, C., Hu, X., Zhang, S., and Zhang, X.: Long-term trend of new particle formation events in the Yangtze River Delta, China and its

influencing factors: 7-year dataset analysis, Science of The Total Environment, 807, 150783, https://doi.org/10.1016/j.scitotenv.2021.150783, 2022.

Sun, T., Che, H., Qi, B., Wang, Y., Dong, Y., Xia, X., Wang, H., Gui, K., Zheng, Y., Zhao, H., Ma, Q., Du, R., and Zhang, X.: Aerosol optical characteristics and their vertical distributions under enhanced haze pollution events: effect of the regional transport of different aerosol types over eastern China, Atmos. Chem. Phys., 18, 2949-2971, 10.5194/acp-18-2949-2018, 2018.

Wang, Y., Niu, S., Lu, C., Lv, J., Zhang, J., Zhang, H., Zhang, S., Shao, N., Sun, W., Jin, Y., and Song, Q.: Observational study of the physical and chemical characteristics of the winter radiation fog in the tropical rainforest in Xishuangbanna, China, Sci. China, Ser. D Earth Sci., 64, 1982-1995, https://doi.org/10.1007/s11430-020-9766-4, 2021.

Wang, Y., Li, J., Fang, F., Zhang, P., He, J., Pöhlker, M. L., Henning, S., Tang, C., Jia, H., Wang, Y., Jian, B., Shi, J., and Huang, J.: In-situ observations reveal weak hygroscopicity in the Southern Tibetan Plateau: implications for aerosol activation and indirect effects, npj Clim. Atmos. Sci., 7, https://doi.org/10.1038/s41612-024-00629-x, 2024.

---

## Author Comment (AC3)

**Response to community' comments on "Characterization of fog microphysics and their relationships with visibility at a mountain site in China"**

**General comment:**

Thank you for your interesting research manuscript! We discussed your work within our research group since we are doing similar research and got interested in your findings. We have some remarks and questions concerning the experimental set-up. Many technical details (e.g. on the sampling efficiency of the GCVI inlet) are currently missing and should be added to allow a reliable assessment of the presented results. Moreover, the reasoning behind many of the key findings are often not clear to the reader and some more clarifications (incl. add the right references) would clearly help here.

[*Response*] Thanks you very much for your interests in our work and the positive comments and suggestions. We have revised the manuscript according to the comments point by point.

Below, we have listed a few questions and remarks.

1. Our most important comment is the lack of describing the sampling efficiency of the GCVI system. Has it been determined? How well do you sample larger droplets? Have zero-measurements been performed? This is an important task and will have a substantial impact on most of the results and interpretation presented here. (see e.g. Figure S4 and others in Karlsson et al, 2021). At the moment, it is not clear if any particle loss corrections (for the aerosol instrumentation behind the inlets and for the fog monitor) have been done.

[*Response*] Thanks for your suggestion. Before the observation, we operated the GCVI inlet system on a clear day for zero-measurements. The CPC installed downstream of the GCVI system and measured a concentration of 0 during this test. We admit the correction of sampling efficiency of GCVI system is quite important for the

quantitation of cloud residual particles. This study mainly focused on the effects of aerosols on fog microphysical characteristics. For the measurements downstream of the GCVI inlet, only the activation ratio of cloud residual particles was shown in Fig. 7 and Fig. S10 to exhibit the scavenging of un-activated aerosol particle by large fog droplets. This activation ratio, defined as the CCN number concentration to the total particle concentration, is almost not influenced by the sampling efficiency in GCVI systems. As it reported in Karlsson et el. (2021), the correction of sampling efficiency plays an important role in comparing the GCVI measurements to other instrumentation methods, such as aerosol particle measurements from the whole-air inlet and fog droplets size distribution measured by fog monitor. Ongoing work will address these descriptions as well as a comparison of the physicochemical properties of cloud residual particles and cloud interstitial particles.

2. A schematic of the set-up which includes instrument names, inlets, piping, flow rates (or a reference to it) would be very useful to the reader.

[*Response*] Thanks for your suggestion. We have added a figure showing the instruments installation in the supplement as below:

[Figure]

GCVI: Ground-based Counterflow Virtual Impactor    AMS:Aerosol Mass Spectrometer
TSMPS: Twin Scanning Mobility Particle Sizer        CCNc: Cloud Condensation Nuclei Counter

Fig. S2. Schematic of the experimental setup at the Daming Mountain site. An automatic three-way switching system was placed between the sample inlets and instruments. Meteorological parameters and fog droplets were simultaneously measured on the roof of the observation container. The bypass pump only operated when the three-way valve connected to the $PM_{2.5}$ inlet. Its flow rate was controlled at 4.5 L min$^{-1}$ via a mass flow controller, ensuring the total sample flow reached the 16.7 L min$^{-1}$ required by the $PM_{2.5}$ cyclone inlet.

3. Line 23 and 240: How do you know that it was indeed collision-coalescence? Just because another peak in the size distribution appeared? Could it be that it is just condensational growth? Please elaborate.

[*Response*] Thanks for pointing this out. In the mature stage, $N_d$ experienced a significant decrease due to a substantial reduction in small droplets, meanwhile, $D_{eff}$ notably increased with an additional peak of the droplets size distribution appearing at 23 μm. Besides that, the activation ratio of fog residual particles significantly reduced during this stage, implying certain un-activated aerosol particles were scavenged by the uptake of larger fog droplets. Based on the evidences described above, we infer that the collision-coalescence process occurred at this stage.

4. One of the key findings is that secondary activation was observed after additional cooling. However, this is not really clear from the figures and key parameters like wind-direction and speed are not shown.

[*Response*] Thanks for your suggestion. The wind direction and speed have been added in Fig. 5 in the revised manuscript. We also added the descriptions on the secondary activation process as below:

"However, after reaching and maintaining a quasi-equilibrium supersaturation state ($SS_{Q1}$) in the early mature stage, a notable decrease in temperature occurred (Fig. 5a) without obvious changes in wind direction and speed (Fig. 5b). This decrease caused an increase in both excess water vapor pressure and supersaturation, as the temperature-dependent equilibrium vapor pressure dropped faster than the ambient partial vapor pressure. Consequently, a new quasi-equilibrium supersaturation state ($SS_{Q2}$) was established, exhibiting distinct fog microphysical characteristics (Fig. 6b). Compared

to $SS_{Q1}$, the $N_d$ substantially increased in the $SS_{Q2}$ stage, while the $LWC$ and $D_{eff}$ notably decreased (Fig. 5b). The enhanced $SS$ facilitated the further activation of smaller particles that were un-activated during the $SS_{Q1}$ stage, resulting in a secondary activation-dominated process during the E3 (Fig. 5c and Fig. 6b)."

5. Introduction: Please refer to e.g. Elias et al. (2009) and Hammer et al. (2014) who also discussed the contribution of hydrated aerosol to light extinction.

[*Response*] Thanks for your suggestion. We add the relevant references in the revised manuscript. Please see Line 73.

6. Line 68: You mention that particle number size distributions were measured but the actual findings/curves (mean distributions and timelines) are never shown. However, it would be useful to add these graphs to the manuscript or SI to better interpret the findings.

[*Response*] Thanks for pointing this out. This study primarily focuses on the evolutionary characteristics of fog microphysical processes. The number size distributions of pre-fog aerosols, fog interstitial particles, and fog residual particles during this campaign are present in another study (Shen et al., 2024), which is also in preprint of ACP.

Shen, X., Liu, Q., Sun, J., Kong, W., Ma, Q., Qi, B., Han, L., Zhang, Y., Liang, L., Liu, L., Liu, S., Hu, X., Lu, J., Yu, A., Che, H., and Zhang, X.: Measurement report: The influence of particle number size distribution and hygroscopicity on the microphysical properties of cloud droplets at a mountain site, EGUsphere, 2024, 1-24, 10.5194/egusphere-2024-2850, 2024.

7. Line 83: Do you have supporting data showing that the site is usually near the top of the PBL?

[*Response*] Thanks for pointing this out. Due to the lack of measurement of the PBL height on this site, we have removed the relevant description from the revised manuscript.

8. Line 132: The CCN counters are usually kept longer at one fixed temperature in

order to achieve a stable supersaturation. Have you checked that 1 min is a long enough period? Especially, when switching from 0.7% down to 0.1% we doubt that this will be sufficient.

[*Response*] We agree your opinion. We had checked the CCN data. After altering the *SS* in the CCNc column, the CCN concentration can reach a stable state within 1 min. In our study, the four *SS* setpoints were sequentially scanned from low to high and then back from high to low to avoid large change of SS in the CCNc column. The relevant descriptions had been added in the revised manuscript. Please see Lines 139-140.

9. Line 149: Consider including the interstitial aerosol number concentration in Tab. 1. Have you performed a closure study to see if $N_d$ and the number of fog residuals agree?

[*Response*] Thanks for your suggestion. In table 1, we give out the microphysical parameters of 8 fog events during the campaign. This manuscript focuses on the analysis of fog monitor data with some support from CCNc and TSMPS data. Ongoing work will address the comparison of fog monitor data and fog residuals data, such as measured $N_d$ and estimated $N_d$, following the method of Karlsson et al. (2021), and will also derive $N_d$ from fog interstitial particles or fog residual particles.

10. Line 176: What is the p-value if you talk about significance but only have a few data points? Have you also looked at the size distribution? Has that also changed in the different pre-fog $N_{total}$ conditions? If you have so much more particles than droplets, why would you expect the $N_{total}$ to be correlated with $N_d$ and not just the $N_{100}$ or even higher? How does the size distribution behind the CVI look like?

[*Response*] Thanks for pointing this out. Indeed, the size distribution of pre-fog aerosols varied in different fog events. We have performed a t-test for the correlation between pre-fog aerosols and the peak $N_d$. The *p*-values for both pre-fog $N_{a\_total}$ and $N_{a\_100}$ were less than 0.05, indicating a significant level of correlation for them. We have added the *p*-values in Fig. 3 and Fig. S5. As shown, the concentrations of particle diameter larger than 70 nm ($N_{a\_70}$) or 100 nm ($N_{a\_100}$) had a much stronger correlation with the peak $N_d$ than that of total pre-fog $N_a$. The particles number size distribution

behind the CVI during this campaign can be found in Shen et al. (2024).

11. Line 180 and Fig. 3: It is not really clear why certain points were excluded. Please explain and reason why the data points after rain events should be excluded. Do you then sample artifacts? In addition, in Fig. 3, please state which kind of linear regression has been applied. Since both x- and y-values are prone to errors, you should use an orthogonal regression, which does not seem to be used here.

[*Response*] Thanks for pointing this out. For the fog events occurred without precipitation, the $N_d$ dramatically increased due to the activation of aerosol particles. This is also the main reason for the good positive correlation between the pre-fog $N_a$ and the peak $N_d$. However, raindrops can significantly influence the $N_d$ through collision-coalescence, resulting in no clear correlation between pre-fog $N_a$ and peak $N_d$. When precipitation was detected by a rain/snow sensor, the GCVI inlet system automatically shut down, and the sampling flow from $PM_{2.5}$ pathway. Additionally, the linear regression used in Fig. 3 is based on the least squares fit, the *p*-values have been added in this figure.

12. Line 184: What does it indicate that your slope values are higher than those measured by Duplessis et al. (2021)? What are the consequences? It would help to elaborate more here.

[*Response*] Thanks for pointing this out. We have added the description to illustrate its implication as below:

"The slope value of 0.09 in this study is significantly higher than the 0.014 observed by Duplessis et al. (2021) on the eastern coast of Canada, indicating stronger bulk activity observed at this mountain site."

13. Line 214: Please also include the *SS* values from those four publications you are referring to.

[*Response*] We have added these *SS* values in the revised manuscript. Please see Lines 265-268.

14. Line 225: To us, the classification into fog stages seems to only have worked semi-well, especially in E3. How exactly did you divide the fog events and why did you choose this definition?

*[Response]* Thanks for pointing this out. The changes in visibility during fog events are closely related to the evolution of fog microphysical characteristics. The classification of fog stages can be based on changes in visibility (Mazoyer et al., 2022) or the ratio of *LWC* to $N_d$ (Li et al., 2020). The classification depending on visibility was adopted not only in reference to previous studies (Mazoyer et al., 2022; Niu et al., 2010b; Pilie et al., 1975), but also to align with the subsequent discussions in this paper, regarding the relationship between fog microphysical parameters and visibility.

15. Line 228: What concentrations do you consider to be "high" or "low"? It would improve the interpretation to explicitly state the measured concentrations here.

*[Response]* Thanks for pointing this out. We have added the descriptions of aerosol backgrounds as below:

"Although there were few anthropogenic sources near the site, the observed aerosol concentrations varied dramatically. As shown in Fig. 1e, the $N_a$ ranged from 230 to 15620 cm$^{-3}$, with a median of 2750 cm$^{-3}$. Episodes with $N_a$ exceeding 8000 cm$^{-3}$ were typically associated with a pronounced increase in aerosol number concentration within the size range of 10-100 nm (Fig. 1e), which were likely driven by new particle formation (Shen et al., 2022). In the subsequent discussion, the pre-fog aerosol concentration below and above this median were defined as low and high number concentrations of aerosol backgrounds, respectively."

16. Line 242ff: Why should the activation ratio of the residuals show that there was scavenging? Please explain in more detail as a correlation doesn't mean causality. What is the reasoning to define the AR via the CCNC measurement and not via the CPC/SMPS measurements?

*[Response]* The estimating water vapor supersaturation (*SS*) in fogs during this

campaign were generally lower than 0.2%. If fog residual particles enter droplet though an activation process, these particles should also be activated in the CCN counter (CCNc) column, where can set different $SS$ conditions. The activation ratio (AR) was defined as the CCN concentration measured by CCNc to the aerosol concentration measured by SMPS. In this case, the concentrations measured by CCNc and TSMPS after GCVI inlet should be consistent, i.e., the AR should be ~1, especially for high $SS$ setpoints. As shown in Fig. 7, the AR measured downstream of the GCVI airflow were closed to 1 when the $D_{eff}$ of fog droplets smaller than 12 μm. However, when the $D_{eff}$ exceeding 12 μm, the AR of fog residual particles notably decreased. The reduced AR of cloud residual particles was caused by the uptake of particles less prone to activation into droplets, implying the fog scavenging efficiency for these particles significantly enhanced in this stage. We add these descriptions and figures of the AR variations under different $SS$ conditions in the revised manuscript. Please see Lines 303-308 and Fig. S10.

17. Line 259ff: The statement about the evaporation of large droplets due to the formation of smaller droplets is not really clear. Could you elaborate more here and also provide some more references for this effect?

*[Response]* Thanks for pointing this out. We are sorry that we did not express clearly in the original manuscript. Because both large and small droplets are affected by evaporation, but small droplets are more susceptible to dry air because of a larger surface area concentration. The description has been revised as below:

"During this secondary activation process, a greater number of small droplets formed and competed for the limited water vapor, which led to a decrease in the $D_{eff}$ (Fig. 6b)."

18. Line 270: Why are you using 880nm? Is it because of the visibility sensor that comes with the GCVI? In that case, it should probably also be 3 and not 3.912 in eq. 5 because of how the visibility sensor is calibrated (see manual of the Belfort visibility sensor).

*[Response]* Thanks for pointing this out. The data from Belfort visibility sensor that comes with the GCVI were lacking in several time periods due to GCVI instrument failure. The visibility data used in this study was from a simultaneously measurement of the forward scattering visibility meter (Model DNQ1, Huayun Inc., China) at 880 nm. The numerator in Eq. 5 should be 3, which is in accordance with the method of visibility meter. We apologize for the incorrect value given in Eq.5 and have corrected it in the revised manuscript.

19. I would suggest moving the first part of chapter 3.5.1 to the methods section. This is not really results.

*[Response]* Thans for your suggestion. We have moved this part to the Method section.

20. Line 289: Were all data points included when performing the linear regression? It would be helpful to add the result to the figure. Is the slope similar if you only include values below e.g. 1km?

*[Response]* Yes, the linear regression in Fig. 8a included all data points. We add the fitting lines to the figure. When we select the data of $VIS_{DSD} \leqslant 1000$ m to perform linear regression, only the slope for $VIS_{GN}$ is similar with that of all data points.

21. Line 290ff: Adding a new parameter (here $N_d$) gives more information and therefore improves the parametrization. Please clarify the last two sentences of this paragraph.

*[Response]* The visibility degradation contributed by fog droplets is determined by fog droplets size distribution. Meanwhile, the fog microphysical parameters of $N_d$, *LWC*, and $D_{eff}$ are derived from the measurement of fog droplets size distribution (Equation 1-3). When both *LWC* and $N_d$ values are given, the information of $D_{eff}$ can also be determined to a large extent. Comparing to the *LWC*-only parameterization, the *LWC*·$N_d$ parameterization can better represent the fog droplets size distribution, and therefore is

expected to be more accurate in fog visibility forecasts. We clarify it in revised manuscript. Please see Lines 338-343.

22. Line 300: Mie theory should be a good prediction for observed visibility. Make your explanation more detailed.

*[Response]* We have added the explanation in Line 330-333:
"Compared to the parameterization schemes of fog visibility, Mie theory incorporates a specific extinction algorithm based on physical processes. Therefore, the fog visibility derived from fog DSD and Mie theory is expected to better reflect actual conditions, which can serve as a reference for fog visibility parameterization."

23. Line 254: Please also mark these quasi-equilibrium states in the temporal evolution shown in Fig. 5.

*[Response]* We have marked the quasi-equilibrium states in the revised manuscript.

**Figures:**

Fig. 1: We would recommend you to choose different colorbars which have a more intuitive and uniform distribution of colors, e.g. 'Blues'. Having white in the middle of the colorspectra is very misleading. https://journals.ametsoc.org/view/journals/bams/96/2/bams-d-13-00155.1.xml

*[Response]* Thans for your suggestion. We have changed the colorbar.

Fig. 2: As you calculate LWC by using D and Nd, isn't the outcome of this figure trivial? Maybe move it to the supplement?

*[Response]* Thans for your suggestion. These three parameters are derived from the observed droplets size distribution and Equations 1-3. We have moved it to the supplement.

Fig. 4: please plot dN/dlogD as commonly used. The x-axis should probably be 'nm' not 'um'. Please write somewhere that this is E3 as you later on talk a lot about this

specific event.

*[Response]* Thanks for pointing this out. This is a schematic of method for deriving water vapor supersaturation (*SS*) in fog. We have revised the figure and move it to the Method section. The 'E3' has been added in the figure caption to specify the event.

Fig. 5: For better comparability, we would suggest to use the same colorbar and axis limits for all events (also Fig. 1 and S5) and use the same axis for Dp and Deff in subplots (c).

*[Response]* Thans for your suggestion. We have revised the colorbars and axis limits in these figures. The different axis used for $D_p$ and $D_{eff}$ can help to clearly exhibit their variations.

Fig. 6: Please explain in the figure caption what $SS_{Q1}$ and $SS_{Q2}$ means. Have you considered plotting one subplot where the 4 average size distributions are plotted on top of each other so that one can more easily see the differences? Why did you choose a linear scale for the diameter (x-axis)? Typos: 'Development', 'Dissipation'

*[Response]* Thans for your suggestions. We have added the explanations for $SS_{Q1}$ and $SS_{Q2}$ in the figure caption. The averaged size distributions at the four stages have been plotted in one figure (Fig. S7). The x-axis is presented on a linear scale to clearly exhibit the variations in the large droplet size range.

Fig. 7: How come that you still measure so many fog residuals even though the effective diameter is smaller than the cut-off of the CVI?

*[Response]* Thanks for pointing this out. The datapoints in Fig. 7 are $N_d$ and $D_{eff}$ measured by the Fog Monitor and the color of these datapoints are activation ratio (AR). The AR, which was defined as CCN number concentration relative the total particle concentration, were measured downstream of the GCVI system. As shown in the droplets size distributions during the fog formation stage (Fig. 6), even when $D_{eff}$ was smaller than the cut-size of CVI, there still exist some large droplets exceeding the cut-size, which can be sampled by GCVI system.

Fig. 8a and eq. 6 don't match. Which one is the correct value for a?

*[Response]* Revised. It should be 0.027.

Fig. 8a: isn't the interesting regime the small visibilities when LWC>0? Why did you choose a linear axis and not like in Fig. 8b a log-axis?

*[Response]* We have added the fitting lines to this figure. When we used a log-axis, the fitting lines overlapped and hard to be distinguished.

Fig. 8: I would recommend to stick to the notation that you introduced earlier: VIS_K, VIS_KN, VIS_G, VIS_GN

*[Response]* Revised.

Fig. S1: typo in the colorbar title: 'Terrain'.

*[Response]* Revised.

Fig. S6 isn't mentioned in the text.

*[Response]* Revised. We have added the figure number in Line 289.

Fig. S8: please use dN/dlogD and plot the xaxis on a log-scale. The labels of the 2 curves have been switched: black is dry, blue ambient

*[Response]* Revised.

Please be consistent with your statistic parameters and linear regressions in all plots (r/r^2, p)

*[Response]* Suggestion adopted.

Please also be consistent with the time on the x-axis in the different figures and do not shrink and stretch the time. It makes it hard to compare the different fog events.

*[Response]* Suggestion adopted.

References:

Duplessis, P., Bhatia, S., Hartery, S., Wheeler, M. J., and Chang, R. Y. W.: Microphysics of aerosol, fog and droplet residuals on the Canadian Atlantic coast, Atmospheric Research, 264, 105859, https://doi.org/10.1016/j.atmosres.2021.105859, 2021.

Elias, T., Haeffelin, M., Drobinski, P., Gomes, L., Rangognio, J., Bergot, T., Chazette, P., Raut, J.-C., and Colomb, M.: Particulate contribution to extinction of visible radiation: Pollution, haze, and fog, Atmospheric Research, 92, 443-454, https://doi.org/10.1016/j.atmosres.2009.01.006, 2009.

Hammer, E., Gysel, M., Roberts, G. C., Elias, T., Hofer, J., Hoyle, C. R., Bukowiecki, N., Dupont, J. C., Burnet, F., Baltensperger, U., and Weingartner, E.: Size-dependent particle activation properties in fog during the ParisFog 2012/13 field campaign, Atmos. Chem. Phys., 14, 10517-10533, 10.5194/acp-14-10517-2014, 2014.

Karlsson, L., Krejci, R., Koike, M., Ebell, K., and Zieger, P.: A long-term study of cloud residuals from low-level Arctic clouds, Atmos. Chem. Phys., 21, 8933-8959, 10.5194/acp-21-8933-2021, 2021.

Li, J., Zhu, C., Chen, H., Zhao, D., Xue, L., Wang, X., Li, H., Liu, P., Liu, J., Zhang, C., Mu, Y., Zhang, W., Zhang, L., Herrmann, H., Li, K., Liu, M., and Chen, J.: The evolution of cloud and aerosol microphysics at the summit of Mt. Tai, China, Atmos. Chem. Phys., 20, 13735-13751, 10.5194/acp-20-13735-2020, 2020.

Mazoyer, M., Burnet, F., and Denjean, C.: Experimental study on the evolution of droplet size distribution during the fog life cycle, Atmos. Chem. Phys., 22, 11305-11321, 10.5194/acp-22-11305-2022, 2022.

Niu, S., Lu, C., Liu, Y., Zhao, L., Lü, J., and Yang, J.: Analysis of the microphysical structure of heavy fog using a droplet spectrometer: A case study, Advances in Atmospheric Sciences, 27, 1259-1275, 10.1007/s00376-010-8192-6, 2010.

Pilie, R. J., Mack, E. J., Kocmond, W. C., Rogers, C. W. C., and Eadie, W.: The Life Cycle of Valley Fog. Part I: Micrometeorological Characteristics, Journal of Applied Meteorology, 14, 347-363, 1975.

Shen, X., Liu, Q., Sun, J., Kong, W., Ma, Q., Qi, B., Han, L., Zhang, Y., Liang, L., Liu, L., Liu, S., Hu, X., Lu, J., Yu, A., Che, H., and Zhang, X.: Measurement report: The influence of particle number size distribution and hygroscopicity on the microphysical properties of cloud droplets at a mountain site, EGUsphere, 2024, 1-24, 10.5194/egusphere-2024-2850, 2024.

Shen, X., Sun, J., Ma, Q., Zhang, Y., Zhong, J., Yue, Y., Xia, C., Hu, X., Zhang, S., and Zhang, X.: Long-term trend of new particle formation events in the Yangtze River Delta, China and its influencing factors: 7-year dataset analysis, Science of The Total Environment, 807, 150783, https://doi.org/10.1016/j.scitotenv.2021.150783, 2022.